# `Para-CFlows`: $C^k$-universal Diffeomorphism Approximators as Superior Neural Surrogates

**Junlong Lyu**[*]
Huawei Noah's Ark Lab
Hong Kong SAR, China
lyujunlong@huawei.com

**Zhitang Chen**[*]
Huawei Noah's Ark Lab
Hong Kong SAR, China
chenzhitang2@huawei.com

**Chang Feng**
Huawei Noah's Ark Lab
China
cunwenjing@huawei.com

**Wenjing Cun**
Huawei Noah's Ark Lab
China
fengchang1@huawei.com

**Shengyu Zhu**
Huawei Noah's Ark Lab
China
zhushengyu@huawei.com

**Yanhui Geng**
Huawei Noah's Ark Lab
Hong Kong SAR, China
geng.yanhui@huawei.com

**Zhijie Xu**
Huawei
China
xuzhijie@huawei.com

**Yongwei Chen**
Huawei
China
chenyongwei@huawei.com

## Abstract

Invertible neural networks based on Coupling Flows (`CFlows`) have various applications such as image synthesis and data compression. The approximation universality for `CFlows` is of paramount importance to ensure the model expressiveness. In this paper, we prove that `CFlows` can approximate any diffeomorphism in $C^k$-norm if its layers can approximate certain single-coordinate transforms. Specifically, we derive that a composition of affine coupling layers and invertible linear transforms achieves this universality. Furthermore, in parametric cases where the diffeomorphism depends on some extra parameters, we prove the corresponding approximation theorems for parametric coupling flows named `Para-CFlows`. In practice, we apply `Para-CFlows` as a neural surrogate model in contextual Bayesian optimization tasks, to demonstrate its superiority over other neural surrogate models in terms of optimization performance and gradient approximations. Code will be avaliable at https://gitee.com/mindspore/models/tree/master/research/bo/paracflow.

## 1 Introduction

Invertible neural networks (INNs) such as coupling flows are firstly introduced as a class of generative models with a tractable likelihood [14, 28, 44], and have shown their usefulness and powerfulness in various machine learning tasks such as inverse problems [2], probabilistic inference [32] and feature extraction [25] in recent years. With plenty of successful applications of INNs, one would wonder if such a type of models have the universal expressiveness. As most generative models mainly concern about the transform between distributions, existing works such as [22, 26] focused on the expressiveness from the distribution perspective. However, the expressiveness from the distribution perspective does not result in the expressiveness from the mapping perspective, as there are a large (or even infinite) number of functions mapping the given source $\mu$ to the given target $\nu$. In many

---

[*]Equal contribution.

applications, knowing the distributional universality is not yet enough. One may be interested in knowing if the optimal transport [45], which finds emerging applications in many fields, e.g., machine learning [35], wireless communication [33] and economics [18], can be approximated by invertible neural networks. Therefore, beyond the distributional universality, it is also important to investigate the universality from the mapping perspective. As INNs are always invertible, it is natural to consider their approximation ability to diffeomorphisms. Diffeomorphism plays an important role in mathematics, physics and engineering domains with applications in fluid dynamics [16], wave propagation [20], robot controls [37] etc. A natural question comes to the surface: can all diffeomorphisms be approximated by INNs? Besides, whether the INNs can approximate the derivatives in the meanwhile is also interesting and important, e.g., it provides theoretical guarantee for black box optimization tasks based on surrogate models when gradients are utilized.

More importantly, in many real-world problems, e.g., 3D Euclidean groups, ODE systems and invertible PDE systems with time $t$ as its parameter, diffeomorphisms are usually described as a parametric type: a parametric diffeomorphism is a function $f(\boldsymbol{y}, \boldsymbol{x}) : \mathbb{R}^{m+d} \to \mathbb{R}^d$ such that for any fixed parameter $\boldsymbol{y}_0 \in \mathbb{R}^m$, $f(\boldsymbol{y}_0, \cdot)$ is a diffeomorphism between $\mathbb{R}^d$ and $\mathbb{R}^d$. It is important to know if they can also be approximated by INNs with additional inputs of parameters. Existing works including [42] have been proposed to investigate the universality of INNs over diffeomorphisms, however it is not able to handle $C^k$-diffeomorphisms when $k = d + 1$ with $d$ as the dimension of the space. In this paper, we address this limitation and more importantly, inspired by the contextualization way proposed in [3], we generalize this structural theorem to the parametric diffeomorphism situation, by concatenating parameters with the variable to be transformed, forming a new subgroup over a higher dimensional space, and provide a complete proof. This structural theorem shows that any compact-supported diffeomorphisms can be decomposed into finitely many compositions of single-coordinate transforms, which we can easily prove to be $C^k$-approximated by, e.g., affine coupling flows together with dimension augmentation ([15], [46]).

Lastly, we suggest using `Para-CFlow`, an affine coupling flow model with dimension augmentation and contextualization, as a superior neural surrogate for black box optimization. The superiority lies in: 1) it uses INN architecture to preserve full rank of the action; 2) it has $C^k$-universality when the target system satisfies certain conditions. Specifically, 2) facilitates much more efficient surrogate and gradient-descent based black box optimization compared to other neural surrogates.

Overall, our contribution is three-fold: 1) We improve the results in [42] to higher-order derivatives with a simpler proof, and give an upper bound on number of layers needed for approximating a certain diffeomorphism; 2) We generalize coupling flows to parametric coupling flows and prove their $C^k$-universal approximation to parametric diffeomorphisms using a novel proving technique; 3) We propose a practical neural network structure of the parametric affine coupling flows and verify the advantage of using this for contextual Bayesian Optimization (BO) tasks.

## 2 Preliminary

Here we introduce some prior knowledge on (parametric) diffeomorphism, INNs and universality, as well as existing works on universality of INNs. As our theoretical results are motivated by and improve upon [42], some of our notations and preliminaries are adopted from them. In what follows, we always assume $k, m, n, d \in \mathbb{N}^+$, where $k$ represents the derivative order, and $m, n, d$ represent dimensionalities for some Euclidean spaces. All vectors are supposed to be row vectors.

### 2.1 $C^k$-diffeomorphism groups on $\mathbb{R}^d$

$C^k$-**diffeomorphisms (group).** Consider an invertible map $f$ from $\mathbb{R}^d$ to $\mathbb{R}^d$. $f$ is said to be a $C^k$-diffeomorphism, if $f$ has up to $k$-th continuous derivatives, and $\det Df(\boldsymbol{x}) \neq 0$ for all $\boldsymbol{x}$. One can easily verify that, given $f, g : C^k$ diffeomorphisms from $\mathbb{R}^d$ to $\mathbb{R}^d$, $f \circ g$ and $f^{-1}$ is still a $C^k$-diffeomorphism from $\mathbb{R}^d$ to $\mathbb{R}^d$. If we denote $\text{Diff}^k(\mathbb{R}^d) \triangleq \{f : f \text{ is a } C^k \text{ diffeomorphism over } \mathbb{R}^d\}$, we see that $\text{Diff}^k(\mathbb{R}^d)$ has a natural group structure with composition as its group operator.

**Compactly supported $C^k$-diffeomorphisms.** In real applications, finite data cannot cover the whole space $\mathbb{R}^d$. Besides, existing approximation theories only guarantee the capability over some bounded compact set $K$. Therefore we consider compactly supported functions or diffeomorphisms. A function $f : \mathbb{R}^d \to \mathbb{R}$ is said to be compactly supported, if there exists a compact set $K$ such

that $f(\boldsymbol{x}) = 0$, for all $\boldsymbol{x} \notin K$. Similarly, a diffeomorphism $f$ is said to be compactly supported, if there exists a compact set $K$ such that $f(\boldsymbol{x}) = \boldsymbol{x}$, for all $\boldsymbol{x} \notin K$, resulting in that all the non-diagonal components of $Df$ are compactly supported. We define $\mathrm{Diff}_c^k(\mathbb{R}^d) \triangleq \{f \in \mathrm{Diff}^k(\mathbb{R}^d) : f \text{ compactly supported}\}$. Obviously $\mathrm{Diff}_c^k(\mathbb{R}^d)$ is a subgroup of $\mathrm{Diff}^k(\mathbb{R}^d)$.

$C^k$**-parametric diffeomorphisms.** A parametric diffeomorphism is a family of diffeomorphisms with some parameter $\alpha$: $\{f_\alpha\}_{\alpha \in A}$, where for any given $\alpha$, $f_\alpha$ is a $\mathbb{R}^d \to \mathbb{R}^d$ diffeomorphism. Usually the parameter is described by some vector $\boldsymbol{y} \in \mathbb{R}^m$, and $f_{\boldsymbol{y}}$ varies continuously or smoothly w.r.t. $\boldsymbol{y}$. We denote $f(\boldsymbol{y}, \boldsymbol{x}) = f_{\boldsymbol{y}}(\boldsymbol{x})$. We can embed it into a higher dimensional diffeomorphism $F(\boldsymbol{y}, \boldsymbol{x}) = \left(\boldsymbol{y}, f(\boldsymbol{y}, \boldsymbol{x})\right)$. One can verify directly that $F \in \mathrm{Diff}^k(\mathbb{R}^{m+d})$ given that (1) $f(\boldsymbol{y}_0, \boldsymbol{x}) \in \mathrm{Diff}^k(\mathbb{R}^d)$ for any fixed $\boldsymbol{y}_0$; (2) $f$ is $k$-th differentiable w.r.t. $(\boldsymbol{y}, \boldsymbol{x})$. Obviously $\mathrm{Diff}_c^{k,m,d} \triangleq \{F \in \mathrm{Diff}_c^k(\mathbb{R}^{m+d}) : F(\boldsymbol{y}, \boldsymbol{x}) = (\boldsymbol{y}, f(\boldsymbol{y}, \boldsymbol{x})) \text{ with } \boldsymbol{y} \in \mathbb{R}^m, \boldsymbol{x} \in \mathbb{R}^d\}$, whose elements keep the first $m$ coordinates and change the last $d$ ones, is a subgroup of $\mathrm{Diff}_c^k(\mathbb{R}^{m+d})$.

## 2.2 INNs based on parametric coupling flows

Here we introduce the classical INNs and investigate the space generated by them.

**Invertible linear transforms.** First, let us define the invertible linear transforms (ILT):

$$\mathrm{ILT}_d \triangleq \{\mathcal{L} : \mathcal{L}\boldsymbol{x}^T = A\boldsymbol{x}^T + \boldsymbol{b}^T, A \in \mathrm{GL}_d(\mathbb{R}), \boldsymbol{b} \in \mathbb{R}^d\}$$

and the parametric case where $\boldsymbol{y}$ are parameters, here $\mathrm{GL}_d(\mathbb{R})$ includes all $d \times d$ regular matrices.

$$\mathrm{ILT}_{m,d} \triangleq \{\mathcal{L} : \mathcal{L}\begin{pmatrix} \boldsymbol{y}^T \\ \boldsymbol{x}^T \end{pmatrix} = \begin{pmatrix} I_m & 0 \\ B & A \end{pmatrix}\begin{pmatrix} \boldsymbol{y}^T \\ \boldsymbol{x}^T \end{pmatrix} + \begin{pmatrix} \boldsymbol{0} \\ \boldsymbol{b}^T \end{pmatrix}, A \in \mathrm{GL}_d(\mathbb{R}), \boldsymbol{b} \in \mathbb{R}^d \ B \in \mathbb{R}^{d \times m}\}.$$

**Coupling flows.** We now define invertible coupling flows [34], a specific type of nonlinear transforms:

$$\Phi_{d,i,\phi} : \mathbb{R}^d \longrightarrow \mathbb{R}^d, \quad (\boldsymbol{x}_{\leq i}, \boldsymbol{x}_{>i}) \longmapsto (\boldsymbol{x}_{\leq i}, \phi(\boldsymbol{x}_{\leq i}, \boldsymbol{x}_{>i})),$$

where $\phi(\boldsymbol{x}_{\leq i}, \cdot) : \mathbb{R}^{d-i} \to \mathbb{R}^{d-i}$ is a diffeomorphism for each fixed $\boldsymbol{x}_{\leq i}$. Specifically, when $\phi(\boldsymbol{x}_{\leq i}, \boldsymbol{x}_{>i}) = \boldsymbol{x}_{>i} \odot \exp\left(\sigma(\boldsymbol{x}_{\leq i})\right) + t(\boldsymbol{x}_{\leq i})$, it is the so-called affine coupling flow [13] and we denote $\Phi_{d,i,\sigma,t} = \Phi_{d,i,\phi}$ for such $\phi$. $\sigma, t$ are some functions with $d - i$ output units, typically modeled with deep neural networks. $\odot$ represents the point-wise product. $\boldsymbol{x}_{\leq i} = (x_1, \cdots, x_i)$, $\boldsymbol{x}_{>i} = (x_{i+1}, \cdots, x_d)$ for $\boldsymbol{x} = (x_1, \cdots, x_d)$. Similarly, for parametric cases, we have:

$$\Phi_{d,i,m,\phi} : \mathbb{R}^{m+d} \longrightarrow \mathbb{R}^{m+d}, \quad (\boldsymbol{y}, \boldsymbol{x}_{\leq i}, \boldsymbol{x}_{>i}) \longmapsto (\boldsymbol{y}, \boldsymbol{x}_{\leq i}, \phi(\boldsymbol{y}, \boldsymbol{x}_{\leq i}, \boldsymbol{x}_{>i})), \tag{1}$$

and specifically, $\phi(\boldsymbol{y}, \boldsymbol{x}_{\leq i}, \boldsymbol{x}_{>i}) = \boldsymbol{x}_{>i} \odot \exp\left(\sigma(\boldsymbol{y}, \boldsymbol{x}_{\leq i})\right) + t(\boldsymbol{y}, \boldsymbol{x}_{\leq i})$ for affine-type coupling flows and we denote $\Phi_{d,i,m,\sigma,t} = \Phi_{d,i,m,\phi}$ for such $\phi$.

**Single-coordinate affine coupling flows** (SACFs) denotes the flows with only the last coordinate changed. Let $\mathcal{H}_d$ be a set of functions from $\mathbb{R}^d$ to $\mathbb{R}$. We define

$$\mathcal{H}\text{-SACF}_d \triangleq \{\Phi_{d,d-1,\sigma,t} : \sigma, t \in \mathcal{H}_{d-1}\}, \text{and } \mathcal{H}\text{-SACF}_{m,d} \triangleq \{\Phi_{d,d-1,m,\sigma,t} : \sigma, t \in \mathcal{H}_{m+d-1}\}.$$

Note that any multi-coordinates affine coupling flows can be represented by finite composition of SACFs and invertible linear transforms, it suffices to just consider the universality of SACFs.

**Invertible neural networks.** Now let us combine linear invertible transform layers and some coupling flow layers to construct our INNs. Let $\mathcal{G}$ be a set consisting of invertible coupling flows. We define the set of INNs based on $\mathcal{G}$ and the parametric case:

$$\mathcal{G}\text{-INN}_d \triangleq \{g_s \circ W_s \circ \cdots \circ g_1 \circ W_1 : s \in \mathbb{N}, g_i \in \mathcal{G}, W_i \in \mathrm{ILT}_d\},$$

$$\mathcal{G}\text{-INN}_{m,d} \triangleq \{g_s \circ W_s \circ \cdots \circ g_1 \circ W_1 : s \in \mathbb{N}, g_i \in \mathcal{G}, W_i \in \mathrm{ILT}_{m,d}\}.$$

When $\mathcal{G}$ contains $\mathcal{H}\text{-SACF}_d$ (or $\mathcal{H}\text{-SACF}_{m,d}$ in parametric cases), it is equivalent to replacing $\mathrm{ILT}_d$ (or $\mathrm{ILT}_{m,d}$) by the symmetric group $S_d$ containing all the permutation over $d$ coordinates operating on $\boldsymbol{x}$. This type of networks are well known as Real-NVPs [13]. One can also define other types of coupling flows. Nevertheless, the theoretical guarantee of any coupling flow can be verified by simply checking their universality to single-coordinate transforms, as our main result stated in Thm. 3.5.

## 2.3 Different types of universality and their relations

Here we define the functional universality. See Appx. A for distributional universality and more.

**Definition 2.1.** *($L^p/L^\infty/C^k$-universality [9]). Let $\mathcal{M}$ be a set of measurable mappings from $\mathbb{R}^n$ to $\mathbb{R}^d$. Let $p \in [1, \infty)$, $k \in \mathbb{N}^+$ and let $\mathcal{F}$ be a set of measurable mappings $f : U_f \to \mathbb{R}^d$, where $U_f$ is a measurable subset of $\mathbb{R}^n$ which may depend on $f$. We say that $\mathcal{M}$ has $L^p$ (or $L^\infty$, $C^k$)-universality for $\mathcal{F}$, if for any $f \in \mathcal{F}$ any $\epsilon > 0$, and any compact subset $K \subseteq U_f$, there exists a $g \in \mathcal{M}$ such that, $\|f - g\|_{L^p(K)} < \epsilon$ (or $\|f - g\|_{L^\infty(K)}, \|f - g\|_{C^k(K)} < \epsilon$, definition of these norms in Appx. A).*

If $\mathcal{M}$ has the $C^k$-universality, $\mathcal{M}$ has the $L^\infty$-universality because $C^k(K)$ is dense in $L^\infty(K)$ for any compact $K$. Also, if $\mathcal{M}$ has the $L^\infty$-universality, $\mathcal{M}$ has the $L^p$-universality for any $1 \le p < \infty$. If an $L^p$-universality is satisfied for some $1 < p < \infty$, then for any $1 \le q < p$, $L^q$-universality is ensured. Finally, $L^1$-universality implies distributional universality but the reverse is not true.

## 2.4 Related works

**Coupling (or triangular) flows.** Authors of [24, 6] proved that the distributional universality of a flow family $h$ can be deduced, if $h$ is dense in the set of all monotone functions by pointwise convergence topology. In particular, [22] proved the distributional universality for neural autoregressive flows, and [26] proved that for sum-of-square flows. [42] generalized it to $L^p$ universality with $1 \le p < \infty$ for affine coupling flows, and $L^\infty$ universality for autoregressive flows and sum-of-square flows.

**Non-coupling flows.** The expressiveness of general non-coupling flows is not well studied, as most existing works restrict the form of nonlinearity to certain types, e.g., planar and radial flows [36], Sylvester flows [43], in order to easily compute the determinant of the Jacobian matirx and the inverse maps. [29] gave a first study over some specific distributions, while the universal distributional results still remains unknown. Another type, iResNet [5], was proposed based on residual network (ResNet) [21] to improve nonlinearity as well as the log determinant computation efficiency. [46] proved that iResNet, capped by linear layers or with extra dimensions, has $C^0$-universality.

**Continuous time flows.** The ODE-based method is also a major class of flow models as introduced in [10, 19, 11, 38]. [11] gave counterexamples for the $C^0$-universality of neural ODEs, however its distributional universality is not yet addressed. An "augmented" neural ODEs was proposed by [15] and then analyzed by [46]. They showed that embedding the original $d$ dimensional neural ODE into a $d + 1$ dimensional space can bypass the original counterexamples, achieving a $C^0$-universality.

# 3 Main results

In this section, we present our main results, following the architecture similar to [42]. Section 3.1 provides a proof for non-parametric cases, showing that: 1) the $C^k$-universality over the $\text{Diff}_c^k(\mathbb{R}^d)$ can be achieved when the $C^k$-universality over a simple subgroup of $\text{Diff}_c^k(\mathbb{R}^d)$ is ensured; 2) affine coupling layers with one zero-padding has $C^k$-universality over such a simple subgroup. In addition, we estimate the number of layers needed for approximate certain diffeomorphism in Thm 3.3. In Section 3.2, we generalize these results to parametric cases with a novel proof technique.

## 3.1 Non-parametric cases

Here we outline the main steps of our proof, and the complete proof is available in Appx. B.

Recall the space $\text{Diff}_c^k(\mathbb{R}^d)$: All the $C^k, \mathbb{R}^d \to \mathbb{R}^d$ diffeomorphisms which are compactly supported. There are two advantages of choosing $\text{Diff}_c^k(\mathbb{R}^d)$ instead of $\text{Diff}^k(\mathbb{R}^d)$: firstly the compactly supported property greatly simplifies the structure of this group; secondly for any diffeomorphism $F \in \text{Diff}^k(\mathbb{R}^d)$ and any compact set $K$, there exists a compactly supported diffeomorphism $f \in \text{Diff}_c^k(\mathbb{R}^d)$ such that $F|_K = f|_K$, i.e., $F(\boldsymbol{x}) = f(\boldsymbol{x})$, for all $\boldsymbol{x} \in K$ (See Lemma 6 in [42]).

Directly proving the universality to a general diffeomorphism is difficult. It is beneficial to decompose the original hard problem into a series of simpler ones as follows.

**Definition 3.1.** *A diffeomorphism $\tau \in \text{Diff}_c^k(\mathbb{R}^d)$ is in single-coordinate transforms $S_c^{k,d}$, if $\tau(\boldsymbol{x}) = (x_1, x_2, \cdots, x_{i-1}, \tau_i(\boldsymbol{x}), x_{i+1}, \cdots, x_d)$, i.e., only one coordinate is altered, and $\tau_i(\boldsymbol{x}) = \tau_i(x_1, \cdots, x_d)$ is a function from $\mathbb{R}^d$ to $\mathbb{R}$ which is monotonic for $x_i$ .*

The theorem as follows shows that, all diffeomorphisms in $\mathrm{Diff}_c^k(\mathbb{R}^d)$ can be constructed from $S_c^{k,d}$.

**Theorem 3.2.** *For any subgroup $H \subseteq Diff_c^k(\mathbb{R}^d)$ s.t. $S_c^{k,d} \subseteq H$, $H = Diff_c^k(\mathbb{R}^d)$. That is, for any $f \in Diff_c^k(\mathbb{R}^d)$, there exists $\tau_1, \tau_2, \cdots, \tau_s \in S_c^{k,d}, s \in \mathbb{N}$ s.t. $f = \tau_s \circ \tau_{s-1} \circ \cdots \circ \tau_1$.*

*Proof.* The following two steps are sketched: 1) If $H \subseteq \mathrm{Diff}_c^k(\mathbb{R}^d)$ and $S_c^{k,d} \subseteq H$, $H$ contains all near-identity diffeomorphisms (Def. B.3). The detail is stated in Cor. B.6. 2) If $H \subseteq \mathrm{Diff}_c^k(\mathbb{R}^d)$ contains all near-identity diffeomorphisms, $H = \mathrm{Diff}_c^k(\mathbb{R}^d)$. The detail is stated in Lem. B.4. $\qquad\square$

We further investigate the number of single-coordinate transforms needed to represent an arbitrary diffeomorphism in Thm. 3.3 as follows. The definition of $\mathrm{dist}_{\mathrm{Diff}_c^1(\mathbb{R}^d)}$ and the proof is provided in Appx. B.2

**Theorem 3.3.** *Given $f \in Diff_c^k(\mathbb{R}^d)$, we denote $dist_{Diff_c^1(\mathbb{R}^d)}(f, I) = \ell$ for some $\ell > 0$, the minimal length of paths between $f$ and the identity map $I$ lying in $Diff_c^1(\mathbb{R}^d)$ with the $C^1$-norm induced metric. Then $f$ can be decomposed into $d(d-1)\ell$ single-coordinate transforms.*

Thm. 3.2 decomposes the original general complex diffeomorphism into finitely-many simple diffeomorphisms. It is natural to ask: if we can approximate all the $S_c^{k,d}$ well, can we also approximate $\mathrm{Diff}_c^k(\mathbb{R}^d)$ well? Thm. 3.4 gives a positive answer to the question.

**Theorem 3.4.** *(Approximation for composition.) Suppose $G$ is a group of diffeomorphisms over $\mathbb{R}^d$. Given a set of diffeomorphisms $F$ over $\mathbb{R}^d$, denote $\mathcal{F}$ as the semigroup generated by $F$. If $G$ has $C^k$-universality for $F$, $G$ has $C^k$-universality for $\mathcal{F}$.*

*Proof.* We only need to prove the following statement: If $G$ has $C^k$-universality for $f_1, f_2 \in \mathcal{F}$, then $G$ has $C^k$-universality for $f_1 \circ f_2 \in \mathcal{F}$. By the property of semigroups, we know that $G$ has $C^k$-universality for the whole $\mathcal{F}$. Details can be found in Appx. B.3. $\qquad\square$

Combining Thm. 3.2 and 3.4, noticing that $S_c^{k,d}$ is inverse-invariant (for any $\tau \in S_c^{k,d}, \tau^{-1} \in S_c^{k,d}$), we can immediately obtain following corollary.

**Corollary 3.5.** *Suppose $G$ is a group of diffeomorphisms over $\mathbb{R}^d$, if $G$ has $C^k$-universality for $S_c^{k,d}$, then $G$ has $C^k$-universality for $Diff_c^k(\mathbb{R}^d)$.*

As stated in Cor. 3.5, any INNs that approximate $S_c^{k,d}$ will eventually approximate $\mathrm{Diff}_c^k(\mathbb{R}^d)$ in $C^k$-norm, which is an extension from the $L^p$ and $L^\infty$ universality stated in [42]. Furthermore, we generalize this result to parametric cases in the next section.

**Affine-Coupling Flows.** Now we only need to prove the universality of $\mathcal{G}\text{-INN}_d$ over $S_c^{k,d}$, when $\mathcal{G}$ contains $\mathcal{H}\text{-SACF}_d$. However, even for such a simple class of diffeomorphisms, $\mathcal{G}\text{-INN}_d$ is not able to approximate it very well: existing works [42] shows that $\mathcal{G}\text{-INN}_d$ has $L^p$ universality over $S_c^{k,d}$. Whereas they gave a negative conjecture for $L^\infty$ universality, let alone the $C^k$-universality.

However, despite the lack of easily computed log-determinant for density evaluation, dimension augmentation has been proven to reach better approximation ability for Neural ODEs [15], [46]. Similarly, when the single-transform is embedded into a higher dimension Euclidean space by padding zeros, the approximation ability becomes much stronger, achieving $C^k$-universality, far beyond $L^p$ universality. We first define the canonical immersion $\iota_d$, and submersion $\pi_d$:

$$\iota_d \colon (x_1, \cdots, x_d) \mapsto (x_1 \cdots, x_d, 0), \quad \pi_d \colon (x_1, \cdots, x_d, x_{d+1}) \mapsto (x_1 \cdots, x_d).$$

Thm. 3.6 says by raising the dimension by 1, we can achieve $C^k$-universality. Proof is in Appx. B.4.

**Theorem 3.6.** *For any compact set $K \subseteq \mathbb{R}^d$, and any $\tau \in S_c^{k,d}$, $\tau \colon (x_1, x_2, \cdots, x_{d-1}, x_d) \longrightarrow (x_1, x_2, \cdots, x_{d-1}, \tau_d(\boldsymbol{x}))$, there exists $\tilde{\tau}$ in $\mathcal{G}\text{-INN}_{d+1}$ with $\mathcal{G} \supseteq \mathcal{H}\text{-SACF}_{d+1}$, such that $\|\tau - \pi_d \circ \tilde{\tau} \circ \iota_d\|_{C^k(K)} < \epsilon$, given that $\mathcal{H}_d$ has $C^k$-universality over functions $K \to \mathbb{R}$.*

## 3.2 Parametric cases

Following Section 3.1, we generalize the results to parametric cases. We use a novel proof technique, by concatenating parameters with the primary input to be transformed, forming a new subgroup over a higher dimensional space, and exploiting the structure of this subgroup.

We denote $S_c^{k,m,d} \triangleq S_c^{k,m+d} \cap \text{Diff}_c^{k,m,d}$, which is the set of transforms acting on a single coordinate of $\boldsymbol{x}$. Recall the definition of compactly supported parametric diffeomorphisms group in Section 2.1, where $\boldsymbol{y}$ acts as parameters. One might raise a question: why not directly use the result in non-parametric case, as all the elements in $\text{Diff}_c^{k,m,d}$ are also in $\text{Diff}_c^k(\mathbb{R}^{m+d})$? Note that the decomposition theorem only ensures that $S_c^{k,m+d}$ can generate $\text{Diff}_c^k(\mathbb{R}^{m+d})$ and then $\text{Diff}_c^{k,m,d}$, it does not guarantee that $S_c^{k,m,d}$ can generate $\text{Diff}_c^{k,m,d}$. Without such a guarantee, we have to use operators that may alter coordinates of both $\boldsymbol{y}$ and $\boldsymbol{x}$, and the coordinates of $\boldsymbol{y}$ in the output layer should be exactly the same as the coordinates of $\boldsymbol{y}$ in the input layer, which restricts the flexibility and increases the approximation difficulty. Thus we derive new dedicated theorems for parametric cases.

**Theorem 3.7.** *For any subgroup $H \subseteq \text{Diff}_c^{k,m,d}$ s.t. $S_c^{k,m,d} \subseteq H$, $H = \text{Diff}_c^{k,m,d}$. That is, for any $f \in \text{Diff}_c^{k,m,d}$, there exists $\tau_1, \tau_2, \cdots, \tau_s \in S_c^{k,m,d}, s \in \mathbb{N}$ s.t. $f = \tau_s \circ \tau_{s-1} \circ \cdots \circ \tau_1$.*

*Proof.* The following two steps are sketched: 1) If $H \subseteq \text{Diff}_c^{k,m,d}$ and $S_c^{k,m,d} \subseteq H$, $H$ contains all near-identity diffeomorphisms (Def. B.3). The detail is stated in Cor. B.6. 2) If $H \subseteq \text{Diff}_c^{k,m,d}$ contains all near-identity diffeomorphisms, $H = \text{Diff}_c^{k,m,d}$ The detail is stated in Lem. B.10. □

Given the above theorem, we easily obtain the $C^k$-universality for $\text{Diff}_c^{k,m,d}$ following Section 3.1.

**Theorem 3.8.** *Suppose $G$ is a group of diffeomorphisms over $\mathbb{R}^{m+d}$. If $G$ has $C^k$-universality for $S_c^{k,m,d}$, then $G$ has $C^k$-universality for $\text{Diff}_c^{k,m,d}$.*

A proof of $\mathcal{G}\text{-INN}_{m,d}$ universality over $S_c^{k,m,d}$, when $\mathcal{G}$ contains $\mathcal{H}\text{-SACF}_{m,d}$, follows Thm. 3.6 in Section 3.1, and thus the detail of the proof is skipped here.

## 4 `Para-CFlows` as superior neural surrogates for black box optimization

In real applications such as control and optimization of telecommunication networks, power grid systems, and electronic systems, we aim to optimize a complicated system under various contexts. Specifically, we are interested in an unknown function $b(\boldsymbol{a}, \boldsymbol{c}) : \mathcal{A} \times \mathcal{C} \to \mathbb{R}$, where $\mathcal{A} \subseteq \mathbb{R}^{d_a}, \mathcal{C} \subseteq \mathbb{R}^{d_c}$ are connected and positive measurable. $\boldsymbol{a}$ is a set of actions one can tune and $\boldsymbol{c}$ is a context vector representing exogenous stimuli. In most cases, we are interested in optimizing the function $b$, i.e. finding $\boldsymbol{a}^* = \arg\max_{\boldsymbol{a}} b(\boldsymbol{c}, \boldsymbol{a})$ given a context $\boldsymbol{c}$. Our task is to quickly find $\boldsymbol{a}^*$ at each $\boldsymbol{c}$ using as few queries as possible. In this paper, we advocate to use `Para-CFlows`, affine coupling flow models with dimension augmentation and parameterization (context information inputted as parameters), to model the relationship between $b$ and $\boldsymbol{a}, \boldsymbol{c}$. Specifically, `Para-CFlows` are constructed as follows:

**Step 1**: We first map $\boldsymbol{a}$ into a higher dimensional space $\mathbb{R}^d$ ($d > d_a$): $\phi_0 : \boldsymbol{a} \mapsto (\boldsymbol{a}, \boldsymbol{a}\boldsymbol{W}) \triangleq \tilde{\boldsymbol{a}}^{(0)}$, where $\boldsymbol{W} \in \mathbb{R}^{d_a \times (d-d_a)}$. By construction, $\text{rank}(\phi_0) = d_a$.

**Step 2**: We further compose $\phi_0$ with a series of invertible affine coupling layers $\phi_i, 1 \leq i \leq N$ operating in $\mathbb{R}^d$ (Eq. (1)). For any $\boldsymbol{x} \in \mathbb{R}^d$, let $d' = \max\{d_a, \lceil \frac{d}{2} \rceil\}$, we define

$$\phi_{\boldsymbol{c},i}(\boldsymbol{x}) = (\boldsymbol{x}_{\leq d'}, \boldsymbol{x}_{>d'} \odot \exp(\sigma_i(\boldsymbol{c}, \boldsymbol{x}_{\leq d'})) + t_i(\boldsymbol{c}, \boldsymbol{x}_{\leq d'})),$$

$\sigma_i, t_i$ are functions with $d - d'$ outputs, implemented by any nonlinear functions like DNNs.

**Step 3**: We write $\tilde{\boldsymbol{a}}^{(i)} = \phi_i P_i(\tilde{\boldsymbol{a}}^{(i-1)}), 1 \leq i \leq N$, where $P_i$ are any fixed random permutation, and

$$\tilde{\boldsymbol{a}}^{(N)} = (\phi_{\boldsymbol{c},N} P_N) \circ \cdots \circ (\phi_{\boldsymbol{c},1} P_1) \circ \phi_0(\boldsymbol{a}) \triangleq \phi(\boldsymbol{c}, \boldsymbol{a}), \quad (2)$$

which is followed by a simple function $\hat{b} : \mathbb{R}^d \to \mathbb{R}$ represented by a simple neural network that shall be trained. Then the whole model is trained by minimizing the Mean Square Error (MSE).

$\min_{\sigma_i, t_i, \hat{b}} \frac{1}{M} \sum_{1 \leq j \leq M} \left( b_j - \hat{b}(\phi(\boldsymbol{c}_j, \boldsymbol{a}_j)) \right)^2$ with $b_j = b(\boldsymbol{c}_j, \boldsymbol{a}_j)$ are trainning samples.

Although it is a straight-forward extension of coupling flows and a similar one was found in [3], our key contribution mainly lie in the theoretical part: for the first time, we prove its $C^k$-universality to

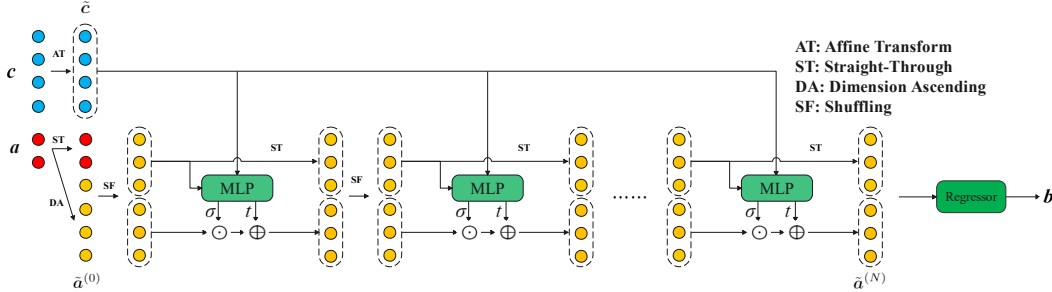

Figure 1: Network structure of `Para-CFlow`

parametric diffeomorphisms and with this property, we advocate to apply it to address an vital action sensitivity issue in black box optimization other than computer vision tasks where flow models are most widely applied.

We now analyze the superiority of `Para-CFlows` as the surrogate for black box optimization.

**Property A: `Para-CFlows` preserves full information of actions.** Existing methods [30, 12, 1, 17, 23] for BO do not handle well the weak action sensitivity issue of many real system optimization tasks [40], where the actions/configuration parameters' effect is overwhelmed by the highly volatile system state and thus the surrogate model $\hat{f}(c, a)$ can easily degenerate to $\hat{f}(c)$ due to a small sample size, especially when $d_c \gg d_a$. On the contrary, `Para-CFlow`, by construction, preserves all the information (the rank) of $a$ up to the last feature extraction layer before the final regression layer.

**Property B: `Para-CFlows` has $C^k$ universality to all rank-preserving systems.** Based on the universality of parametric coupling flows in Section 3, we further investigate what kind of objective function $b$ can be approximated by `Para-CFlows`. We give the necessary condition to $b$ as follows.

**Assumption 4.1.** *$b$ is an observation of the inner system state $\Phi$: $b(c, a) = l(\Phi(c, a))$ and the system is `rank-preserving`, i.e. the inner state $\Phi(c, a) : \mathcal{C} \times \mathcal{A} \to \mathbb{R}^{d_\phi}$ is a differentiable function and preserves full information of actions, i.e., $rank\left(\frac{\partial \Phi}{\partial a}\right) = d_a$.*

Under Assumption 4.1, we can get the following result, which ensures the expressiveness of our architecture,

**Theorem 4.2.** *For any system $\Phi : \mathcal{C} \times \mathcal{A} \to \mathbb{R}^{d_\phi}$ satisfying Asmp. 4.1, if $d \geq 2d_\phi$ and $\Phi$ have up to $k$-th order derivatives, then for any $\epsilon > 0$, we can always find a $\phi$ in the form (2), such that $\|\Phi(c, a) - \pi\phi(c, a)\|_{C^k(\mathcal{C} \times \mathcal{A})} \leq \epsilon$, given that $\sigma_i, t_i, \hat{b}$ are $C^k$-universal function approximator. Here $\pi$ is the canonical submersion: $\mathbb{R}^d \to \mathbb{R}^{d_\phi}$, $(x_1, x_2, \cdots, x_{d_\phi}, \cdots, x_d) \mapsto (x_1, x_2, \cdots, x_{d_\phi})$.*

Readers can check Appx. B.6 for the proof. Thm 4.2 shows that the inner system $\Phi$ can be uniformly approximated by the feature extraction layer $\phi$ of `Para-CFlow`. The observation function $l(\cdot)$ can then be approximated by $\hat{b}$ as a result of the universality of Multi-Layer Perceptons (MLPs).

The $C^k$-approximability property of `Para-CFlow` are of vital importance for black box optimization. Let us just focus on the exploitation phase, i.e., $a^* = \arg\max \hat{b}(\phi(c, a))$. When the action space is continuous and high dimensional (though in this paper, we focus on the context dominating tasks where $d_c \gg d_a$), one is not able to find the optima easily by brute-force search. Many existing works utilize meta-heuristic such as Evolutionary Algorithm (EA) of which the performance heavily depends on the population size and the number of iterations, and thus it is computation demanding and time consuming. Instead, one can apply Gradient Descent (GD) to the surrogate if it is differentiable, i.e. $a^{t+1} \leftarrow a^t + \eta\nabla_a\hat{b}(\phi(c, a))|_{a=a^t}$. However, for the GD method to work, it requires that the surrogate approximates not only the function value but also its first-order derivative. However function value approximation does not necessarily guarantee derivative approximation. A simple illustrative example is $f_n(x) = g(x) + \sin(nx)/n$. One can easily observe that $f_n(x) \to g(x)$, but $f'_n(x) = g'(x) + \cos(nx)$ will never converge to $g'(x)$ when $n \to \infty$. With this argument, we advocate `Para-CFlow` as an adequate choice for surrogate modeling for continuous action space BO. We also find it suitable to approximate Q-function in Reinforcement Learning (RL), especially

for DPG [39], DDPG[31] and references therein where one need to estimate the gradient of the Q-function w.r.t. the action to train the actor network. Extension to DRL will be left for future work.

With the aforementioned properties, our proposed `Para-CFlow` model can serve as a superior neural surrogates and in the setting of BO, we can easily adopt existing SOTA methods of Bayesian deep learning and deep ensembles to facilitate the uncertainty modeling with `Para-CFlow`.

# 5 Experiments

In this section, we conduct experiments to verify the expressiveness of our proposed `Para-CFlow` and demonstrate its advantage as neural surrogate models in application to black-box optimizations.

## 5.1 Learning diffeomorphism "Taiji"

We demonstrate the expressiveness of our model to learn a parametric diffeomorphism with a synthetic task called "Taiji". We define our target parametric diffeomorphism:

$$f_y : (\rho \cos\theta, \rho\sin\theta) \mapsto (\rho\cos\tilde\theta, \rho\sin\tilde\theta), \text{ where } \tilde\theta = \theta + y\arccos(\min(\rho, 1)). \qquad (3)$$

On the unit circle $\rho = 1$, it is not differentiable, thus we expect the trained model to have irregular derivatives over the unit circle, while maintain accurate derivatives elsewhere.

First, we design a task with sufficient amount of data to verify the $C^1$ approximation ability of `Para-CFlows`. We generate samples $\{(\boldsymbol{x}_i, y_i)\}_{1\le i\le 30000}$ with $\boldsymbol{x}_i \sim \mathcal{U}[-1,1]^2$ and $y_i \sim \mathcal{U}[0,1]$ independently. Under the polar coordinate representation, i.e., $\boldsymbol{x}_i = (\rho_i\cos\theta_i, \rho_i\sin\theta_i)$, we calculate the target $f_{y_i}(\boldsymbol{x}_i) = (\rho_i\cos\tilde\theta_i, \rho_i\sin\tilde\theta_i)$ according to Eq. (3). We use a 6-layers affine coupling flows, each layer composed with a random permute. The coupling functions $\sigma, t$ are implemented by 1-hidden-layer Multi-Layer Perceptron (`MLP`) with hidden-unit number comparable to input dimension. We pad zeros to the input $\boldsymbol{x}$, and use the first two output dimensions to compute the loss.

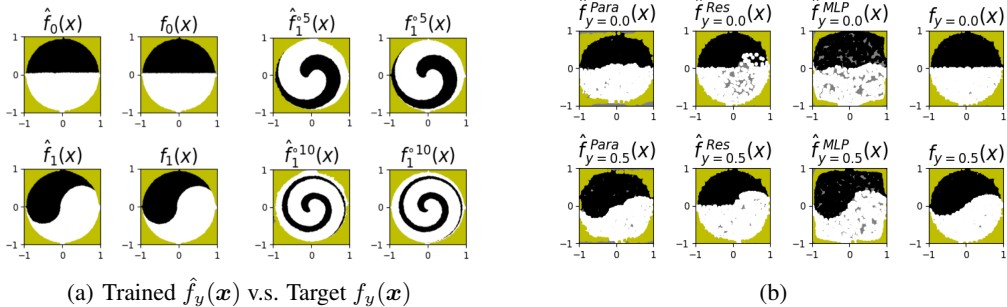

(a) Trained $\hat{f}_y(\boldsymbol{x})$ v.s. Target $f_y(\boldsymbol{x})$               (b)

Figure 2: 2(a): learning "Taiji" diffeomorphism tasks in Section 5.1. Black region, $x_2 > 0 \cap \rho \le 1$; white region, $x_2 \le 0 \cap \rho \le 1$; yellow region, $\rho > 1$. $\rho = \sqrt{x_1^2 + x_2^2}$ and $\boldsymbol{x} = (x_1, x_2)$. $\hat{f}_1^{\circ N}$ denotes the $N$-fold composition $\hat{f}_1 \circ \cdots \circ \hat{f}_1$. 2(b): Comparing `Para-CFlow` with `Resnet` and `MLP` with small sample size and parameter $y^{(i)} = y, 1 \le i \le 100$. Grey color area has no point resides.

On the left panel of Fig. 2(a), we observe good consistency between the trained model $\hat{f}_y(\boldsymbol{x})$ and $f_y(\boldsymbol{x})$, when $y \in [0, 1]$. Note that we only use samples with $y_i \in [0, 1]$ for training, and thus the model generalizability to $y > 1$ is not guaranteed. However, as the true diffeomorphism satisfies $f_{y_1} \circ f_{y_2} = f_{y_1+y_2}$, we iterate the trained model recurrently and expect the final output to achieve the effect for $y > 1$. The results on the right panel of Fig. 2(a) justify that the cumulative error from composition does not grow fast, and our model can approximate complex diffeomorphisms with sufficient layers. In Fig. 3(a) and Fig. 3(b) we plot the derivatives of `Para-CFlow`, compared to the true one. The non-differentiability of the groundtruth on the unit circle results in irregular behaviors of trained model near the unit circle; elsewhere it shows good consistency.

To further verify our theorem stated in Section 3, specifically the effect of padding zero(s), we test the same model under different hidden dimensions $d_{hid}$ and number of stacking layers in Fig. 3(c).

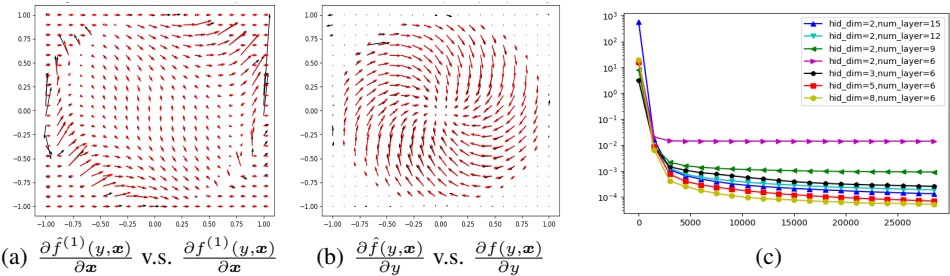

$$\text{(a)} \quad \frac{\partial \hat{f}^{(1)}(y,\boldsymbol{x})}{\partial \boldsymbol{x}} \text{ v.s. } \frac{\partial f^{(1)}(y,\boldsymbol{x})}{\partial \boldsymbol{x}} \qquad \text{(b)} \quad \frac{\partial \hat{f}(y,\boldsymbol{x})}{\partial y} \text{ v.s. } \frac{\partial f(y,\boldsymbol{x})}{\partial y} \qquad \text{(c)}$$

Figure 3: 3(a) and 3(b): Derivatives of the trained model $\hat{f}_y(\boldsymbol{x}) = (\hat{f}^{(1)}(y,\boldsymbol{x}), \hat{f}^{(2)}(y,\boldsymbol{x}))$ with black arrows and the groundtruth $f_y(\boldsymbol{x}) = (f^{(1)}(y,\boldsymbol{x}), f^{(2)}(y,\boldsymbol{x}))$ with red arrows, all at $y = 1$. 3(c): MSE under different hidden dimensions and number of layers.

Here hidden dimension equals to the input dimension (2 in this experiment) plus the number of padding zero(s). As we expected, according to our theorems, with padding zero(s) `Para-CFlow` enjoys universality. We also investigate the expressiveness when hidden dimensions = 2. When we increase the number of stacking layers, MSE also decrease, which justifies the $L^p$ universality result in [42]. However, increasing the hidden dimension is much more efficient in terms of improving expressiveness than increasing the number of stacking layers. Such results support our theory of zero-padding. More results of different architecture settings are available in Appx. D.1.

Next we design a more challenging task to compare `Para-CFlow` against a simple `MLP` with a hidden-layer-setting as $(128, 64, 32)$, and Residual Nets (`Resnet`) [21], that concatenates $\boldsymbol{y}$ and $\boldsymbol{x}$ as the network's input and then appends $\boldsymbol{x}$ to the output of each layer forward for prediction. Unlike the previous experiment where a scalar $y$ is used, we generate 3000 samples of a 100-dimensional parameter vector $\boldsymbol{y}_i = (y_i^{(1)}, \cdots, y_i^{(100)})$ where $y_i^{(j)} \sim \mathcal{N}(y_i, 0.16)$ and $y_i \sim \mathcal{U}[0, 1]$. Then $(\boldsymbol{x}_i, \boldsymbol{y}_i)$ is fed to each model to learn the target $f_{\text{avg}(\boldsymbol{y}_i)}(\boldsymbol{x}_i)$. Fig. 2(b) shows that `Para-CFlow` learn a much more continuous mapping than `MLP` and `Resnet`. While `Resnet` learns relatively better the outer part than `MLP`, similar to `MLP` it suffers from bad mapping in the inner part, i.e. the grey area inside the unit circle is much larger than `Para-Flow`, which represents bad continuity of the learnt mapping.

## 5.2 Application to contextual BO

In this section,we ensemble `Para-CFlows` as a novel surrogate model for BO with three well-known benchmark functions: `Rastrigin`, `Ackley` and `Trid` [41]. Note that the original benchmark tasks are **contextless**, here we simply set the first $d_c$ dimensions as the context vector sampled from a multivariate uniform distribution and leave the last dimension as the action to optimize. To construct context-dominating tasks we set $d_c \gg 1$. In all benchmarks above, the contexts $\boldsymbol{c}$ are uniformly sampled at random from $[-3, 3]^{d_c}$ sequentially while the action $a$ is optimized by gradient descent in $[-3, 3]$ over surrogate models. The used ensemble-like surrogate models for contextual BO are listed as follows: (1) `Para-CFlow`: Ensemble of `Para-CFlows` for prediction with uncertainty; (2) `MLP`: Ensemble of `MLPs` for uncertainty modelling by concatenating context $\boldsymbol{c}$ and $a$ to obtain input $(\boldsymbol{c}, a) \in \mathbb{R}^{d_c+1}$; (3) `MLP-Ascend`: Similar to `MLP`, the difference is that it firstly raises both $\boldsymbol{c}$ and $a$ to the same hidden dimension $\max(5, d_c)$ and then concatenates them as inputs to a `MLP`; (4) `Resnet`: Ensemble of multiple `Resnet` that concatenates $\boldsymbol{c}$ and $a$ as the network's input and then appends $a$ to the output of each layer forward for prediction. More experimental settings are available in Appx. D.

Firstly, we investigate the action sensitivity of each model trained with 100 random samples. The dimensionality of context is set to 5, 10, 20, which is significantly greater than that of action. The action sensitivity is calculated as the ratio of Permutation Feature Importance (`PFI`) [8] of the action w.r.t. that of the context, i.e. $\text{PFI}(a)/\max_i(\text{PFI}(c_i))$ and reported in Tab.1. We observe that `Para-CFlow` and `Resnet` have much higher action sensitivities thanks to the model architecture, while `MLP` and `MLP-Ascend` are dominated by contexts as expected. We further investigate whether different surrogates and their gradients manage to guide the real objective function optimization. Denote each trajectory as $\{(b^t, \hat{b}^t, a^t)_{t=1}^T | \boldsymbol{c}\}$, where $(a^t)_{t=1}^T$ is a $T$-step gradient trajectory over the

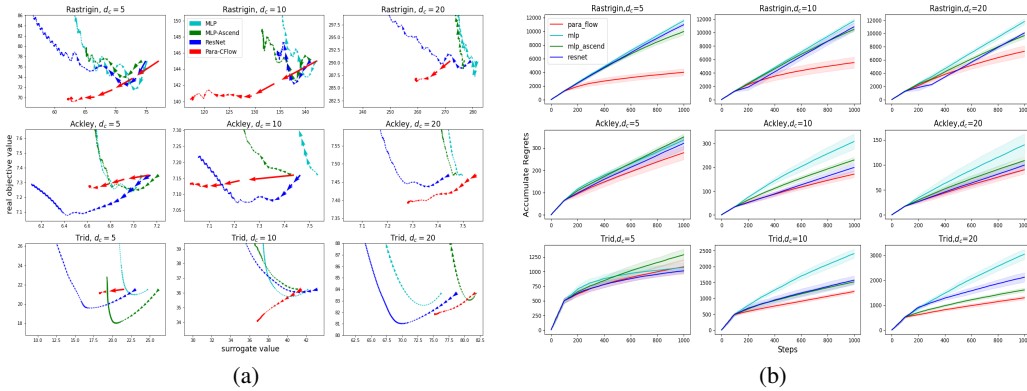

|     |     |
| :-: | :-: |
| (a) | (b) |

Figure 4: 4(a): `quiver plot` of $(\hat{b}^t)_{t=1}^T$ ($x$-axis) v.s. $(b^t)_{t=1}^T$ ($y$-axis) along each averaged gradient trajectory over 10 random initialization $\times$ 10 random contexts. 4(b):The mean and standard deviation of the cumulative regret under 10 independent trials with the context dimension as 5, 10 and 20.

surrogate $\hat{b}(\boldsymbol{c}, a)$ and $b(\boldsymbol{c}, a)$ is the corresponding real objective value . We visualize the `quiver plot` of $(\hat{b}^t)_{t=1}^T$ ($x$-axis) v.s. $(b^t)_{t=1}^T$ ($y$-axis) along each gradient trajectory for all compared surrogates in Fig.4(a). We can easily observe that `Para-CFlow` outperforms other surrogate models as the red arrows move towards the bottom left corner which exactly means the gradient trajectory on `Para-CFlow` indeed minimizes the real objective function, and thus the result supports our main results of $C^k$ approximability, whereas the compared trajectories move towards the top left corner after a few steps probably because those surrogates suffer from more severe local oscillation w.r.t. $a$, comparing to `Para-CFlow`, resulting in unexpected gradient explosion. We then compare different surrogate models' performances on BO. The mean and standard deviation of the cumulative regrets for different surrogate models are reported in Fig. 4(b) where we observe improvement using `Para-CFlow` as the neural surrogates, especially for the `Rastrigin` benchmark.

Table 1: Comparison of action sensitivity over 10 independent trails

|     | Rastrigin | | | Ackley | | | Trid | | |
| --- | --- | --- | --- | --- | --- | --- | --- | --- | --- |
|     | $d_c = 5$ | $d_c = 10$ | $d_c = 20$ | $d_c = 5$ | $d_c = 10$ | $d_c = 20$ | $d_c = 5$ | $d_c = 10$ | $d_c = 20$ |
| **Para-CFlow** | **2.62±0.71** | **4.30±1.16** | **7.14±1.49** | **3.62±1.21** | **4.47±0.95** | **4.37±1.64** | **1.61±0.44** | 1.10±0.51 | 1.34±0.68 |
| **MLP** | 0.89±0.16 | 0.70±0.14 | 0.71±0.13 | 0.91±0.17 | 0.79±0.15 | 0.51±0.16 | 0.71±0.20 | 0.64±0.19 | 0.47±0.20 |
| **MLP-Ascend** | 0.90±0.24 | 0.80±0.21 | 0.15±0.16 | 0.95±0.20 | 0.88±0.10 | 0.58±0.14 | 0.81±0.29 | 0.64±0.19 | 0.19±0.15 |
| **Resnet** | 1.65±0.22 | 1.60±0.22 | 2.32±0.34 | 2.03±0.38 | 1.88±0.25 | 3.44±1.02 | 1.29±0.32 | **1.77±0.39** | **1.74±0.56** |

## 6  Conclusion

In this paper, we firstly prove the equivalence between the $C^k$-universality over compactly-supported diffeomorphisms and that over single-coordinate transforms, resulting in the $C^k$ universal approximation ability of affine coupling flows. Furthermore, we generalize the main theorems to parametric cases and propose a practical model called `Para-CFlows` which could serve as a good neural surrogate for gradient-based search and optimization. With good capabilities both in universal approximation and robust sensing of critical features in parametric diffeomorphisms, we empirically exhibit the advantages of `Para-CFlow` using various benchmarks. However, due to the dimension augmentation (for expressiveness purpose), `Para-CFlow` is not able to benefit from fast log-determinant estimation if it is applied to generative model tasks which will be studied in our future work.

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
