# A Different functional norms and corresponding function spaces

Here we introduce some preliminary definitions on function norms and functions spaces involved in this paper, and the definition for distributional universality.

For a measurable mapping $f : \mathbb{R}^n \to \mathbb{R}^d$ and a subset $K \subseteq \mathbb{R}^n$, we define:

$$\|f\|_{L^p(K)} \triangleq \left( \int_K \|f(\boldsymbol{x})\|^p d\boldsymbol{x} \right)^{1/p}, 1 \leq p < \infty,$$

$$\|f\|_{L^\infty(K)} \triangleq \lim_{p \to \infty} \|f\|_{L^p(K)} \xrightarrow{f \text{ cont.}} \sup_{\boldsymbol{x} \in K} \|f(\boldsymbol{x})\|,$$

where $\|\cdot\|$ can be any norm on $\mathbb{R}^d$, as norms on the finite-dimensional vector space are all equivalent. For simplicity, we choose the maximum norm on $\mathbb{R}^d$, i.e., $\|(x_1, x_2, \cdots, x_d)\| = \max_{1 \leq i \leq d} |x_i|$.

We can also consider the norm containing derivative information when $f$ is $k$-th differentiable:

$$\|f\|_{C^k(K)} \triangleq \sum_{|\boldsymbol{\alpha}| \leq k} \|D^{\boldsymbol{\alpha}} f\|_{L^\infty(K)},$$

where $\boldsymbol{\alpha} = (\alpha_1, \alpha_2, \cdots, \alpha_d) \in \mathbb{N}^d$, $|\boldsymbol{\alpha}| = \sum_{i=1}^d \alpha_i$ and $D^{\boldsymbol{\alpha}} f = \frac{\partial^{|\boldsymbol{\alpha}|} f}{\partial x_1^{\alpha_1} \cdots \partial x_d^{\alpha_d}}$.

With these norms, we can define corresponding function spaces:

$$L^p(K) \triangleq \{\text{Domain}(f) = K : \|f\|_{L^p(K)} < \infty\}$$
$$L^\infty(K) \triangleq \{\text{Domain}(f) = K : \|f\|_{L^\infty(K)} < \infty\}$$
$$C^k(K) \triangleq \{\text{Domain}(f) = K : \|f\|_{C^k(K)} < \infty\}$$

when $K$ is compact, $C^k(K) \subseteq L^\infty(K) \subseteq L^p(K)$, and an important fact is that $C^k(K)$ is dense in $L^p(K)$ under $L^p$ norm. Thus the universality over $C^k(K)$ in $C^k$-norm is stronger than the universality over $L^\infty(K)$ in $L^\infty$ norm, further stronger than the universality over $L^p(K)$ in $L^p$ norm.

**Definition A.1.** *(Distributional universality). Let $\mathcal{M}$ be a set a set of measurable mappings from $\mathbb{R}^n$ to $\mathbb{R}^d$. We say that $\mathcal{M}$ is a distributional universal approximator if for any absolutely continuous probability measure $\mu$ over $\mathbb{R}^n$ w.r.t. Lebesgue measure, and any probability measure $\nu$ over $\mathbb{R}^d$, there exists a sequence $\{g_i\}_{i=1}^\infty \subseteq \mathcal{M}$ such that $(g_i)_* \mu$ converges to $\nu$ in distribution as $i \to \infty$, where $(g_i)_* \mu(A) \triangleq \mu\left(g_i^{-1}(A)\right)$ for any measurable set $A$.*

# B Proofs

## B.1 Proofs for Theorem 3.2

**Definition B.1.** *Isotopies. An isotopy between two diffeomorphisms $\phi_0, \phi_1 \in \text{Diff}_c^k\left(\mathbb{R}^d\right)$ is a $C^k$-map $H : [0, 1] \times \mathbb{R}^d \to \mathbb{R}^d$ such that the mapping $h_t : \mathbb{R}^d \to \mathbb{R}^d$ defined by $h_t(x) = H(t, x)$ for all $t \in [0, 1]$ satisfies $h_0 = \phi_0, h_1 = \phi_1$ and $h_t \in \text{Diff}_c^k\left(\mathbb{R}^d\right)$ for all $t \in [0, 1]$. It turns out that $t \to h_t$ is a continuous path in the group $\text{Diff}_c^k\left(\mathbb{R}^d\right)$ joining $\phi_0$ to $\phi_1$.*

**Proposition B.2.** *(Proposition 1.2.1 in [4]) The group $\text{Diff}_c^k\left(\mathbb{R}^d\right)$ is connected. Moreover, the group $\text{Iso}^k\left(\mathbb{R}^d\right)$ of diffeomorphisms with compact supports which are isotopic to the identity map $I$ through isotopies coincide with $\text{Diff}_c^k\left(\mathbb{R}^d\right)$. Here the identity map $I$ means $I(x) = x$ for all $x \in \mathbb{R}^d$.*

**Definition B.3.** *$(\delta, k)$-near-identity $C^k$-diffeomorphisms. Let $B_{\delta,k}$ be the $C^k$-norm ball with radius $\delta$ and centered at identity map $I(\boldsymbol{x}) = \boldsymbol{x}$, that is to say, $B_{\delta,k} = \{f \in \text{Diff}_c^k\left(\mathbb{R}^d\right) : \sup_{|\alpha| \leq k} \|D^{|\alpha|}(f - I)\|_{L^\infty} < \delta\}$. A diffeomorphism $\phi \in \text{Diff}_c^k\left(\mathbb{R}^d\right)$ is said to be $(\delta, k)$-near-identity, if $\phi \in B_{\delta,k}$.*

**Lemma B.4.** *(Lemma 2.1.8 in [4]) For any diffeomorphism $f \in \text{Diff}_c^k\left(\mathbb{R}^d\right)$ and any $\delta > 0$, there exists a finite sequence of $(\delta, k)$-near-identity diffeomorphisms $g_1, \cdots, g_s$ such that $f = g_s \circ g_{s-1} \circ \cdots \circ g_1$.*

*Proof.* Note that there exists an isotopy $h_t$ from $I$ to $f$ such that $h_0 = I$ and $h_1 = f$. We rewrite $f = h_1 = \left( h_1 \circ h_{(s-1)/s}^{-1} \right) \circ \left( h_{(s-1)/s} \circ h_{(s-2)/s}^{-1} \right) \circ \cdots \circ \left( h_{1/s} \circ h_0^{-1} \right)$ and let $g_i = h_{i/s} \circ h_{(i-1)/s}^{-1}$, we can see that $f = g_s \circ g_{s-1} \circ \cdots \circ g_1$. Take $s$ large enough, we can make $h_{i/s}$ and $h_{(i-1)/s}$ close enough such that $h_{i/s} \circ h_{(i-1)/s}^{-1}$ is $(\delta, k)$-near-identity. $\qquad\square$

**Theorem B.5.** *There exists a $\delta_0 > 0$, such that for any $\delta < \delta_0$ and any $f \in Diff_c^1\left(\mathbb{R}^d\right)$ that is $(\delta, 1)$-near-identity, $f$ can be written as $g \circ h$ with $h(\boldsymbol{x}, y) = \left( \boldsymbol{x}, \tilde{h}(\boldsymbol{x}, y) \right)$ and $g(\boldsymbol{x}, y) = (\tilde{g}(\boldsymbol{x}, y), y)$ for $\boldsymbol{x} \in \mathbb{R}^{d-1}, y \in \mathbb{R}$. If $f \in Diff_c^k\left(\mathbb{R}^d\right)$, so are $g$ and $h$. Further more, $g$ satisfies $\left( \tilde{\delta}, 1 \right)$-near-identity for $\tilde{\delta} = \frac{\delta}{1-\delta} > 0$.*

*Proof.* Let $\pi_i : \mathbb{R}^d \to \mathbb{R}$ denote the projection onto the $i^{th}$ coordinate. Suppose $f : \mathbb{R}^d \to \mathbb{R}^d$ is compactly supported and sufficiently $C^k$-close to the identity. Then for any point $(\boldsymbol{x}, y) = (x_1, \cdots, x_{d-1}, y)$, the map $f_{\boldsymbol{x}} : \mathbb{R} \to \mathbb{R}$ given by $f_{\boldsymbol{x}}(y) = \pi_n f(\boldsymbol{x}, y)$ is a diffeomorphism: surjectivity follows from the fact that $f$ has compact support, which means $\lim_{y \to \pm\infty} f_{\boldsymbol{x}}(y) = \pm\infty$ and by the continuity of $f_{\boldsymbol{x}}$; injectivity follows from the fact that if $f_{\boldsymbol{x}}(y_1) = f_{\boldsymbol{x}}(y_2)$ for some $y_1 \neq y_2$, then $f_{\boldsymbol{x}}$ must has zero derivatives at some point $y \in (y_1, y_2)$, but the derivative of $f_{\boldsymbol{x}}$ respect to $y$ is near 1, a contradiction.

Now given $f$, define $h$ and $g : \mathbb{R}^{d-1} \times \mathbb{R} \to \mathbb{R}^{d-1} \times \mathbb{R}$ by

$$h(\boldsymbol{x}, y) = (\boldsymbol{x}, f_{\boldsymbol{x}}(y)), \text{ and}$$
$$g(\boldsymbol{x}, y) = (g_1(\boldsymbol{x}, y), g_2(\boldsymbol{x}, y), \cdots, g_{d-1}(\boldsymbol{x}, y), y),$$

where $g_i(\boldsymbol{x}, y) = \pi_i\left( f\left( \boldsymbol{x}, f_{\boldsymbol{x}}^{-1}(y) \right) \right) \in \mathbb{R}$. Obviously $g, h \in Diff_c^k\left(\mathbb{R}^d\right)$ given $f \in Diff_c^k\left(\mathbb{R}^d\right)$ and $f = g \circ h$. Also we observe that, $f$ is $(\delta, 1)$-near-identity, thus

$$\sup_{\boldsymbol{x}, y} |\frac{\partial}{\partial y} f_{\boldsymbol{x}}(y) - 1| < \delta, \quad \sup_{\boldsymbol{x}, y} |\frac{\partial}{\partial x_i} f_{\boldsymbol{x}}(y)| < \delta,$$

$$\sup_{\boldsymbol{x}, y} |\frac{\partial}{\partial y} f_{\boldsymbol{x}}^{-1}(y)| = \sup_{\boldsymbol{x}, y} |\frac{\partial}{\partial y} f_{\boldsymbol{x}}(y)|^{-1} < \frac{1}{1-\delta}.$$

Then we have

$$0 = \frac{d}{dx_i} y = \frac{d}{dx_i} \left( f_{\boldsymbol{x}}^{-1}(f_{\boldsymbol{x}}(y)) \right)$$
$$= (\frac{\partial}{\partial x_i} f_{\boldsymbol{x}}^{-1})(f_{\boldsymbol{x}}(y)) + (\frac{\partial}{\partial y} f_{\boldsymbol{x}}^{-1})(f_{\boldsymbol{x}}(y)) \cdot \left( \frac{\partial}{\partial x_i} f_{\boldsymbol{x}}(y) \right),$$

and further

$$|\frac{\partial}{\partial x_i} f_{\boldsymbol{x}}^{-1}(y)| = | \left( \frac{\partial}{\partial y} f_{\boldsymbol{x}}^{-1} \right)(y) \cdot \left( \left( \frac{\partial}{\partial x_i} f_{\boldsymbol{x}} \right)(f_{\boldsymbol{x}}^{-1}(y)) \right) |$$
$$\leq \sup |\frac{\partial}{\partial y} f_{\boldsymbol{x}}^{-1}| \cdot \sup |\frac{\partial}{\partial x_i} f_{\boldsymbol{x}}| < \frac{\delta}{1-\delta}.$$

If we denote $f_j(\boldsymbol{x}, y) = \pi_j f(\boldsymbol{x}, y)$, we get

$$|\frac{d}{dx_i} g_j(\boldsymbol{x}, y) - \delta_{i,j}|$$
$$= |\frac{\partial}{\partial x_i} f_j(\boldsymbol{x}, f_{\boldsymbol{x}}^{-1}(y)) - \delta_{i,j} + \frac{\partial}{\partial y}(f_j(\boldsymbol{x}, f_{\boldsymbol{x}}^{-1}(y))) \cdot \frac{\partial}{\partial x_i} f_{\boldsymbol{x}}^{-1}(y)|$$
$$< \delta + \delta \cdot \frac{\delta}{1-\delta} = \frac{\delta}{1-\delta},$$

which proved that $g$ is $\left( \frac{\delta}{1-\delta}, 1 \right)$-near-identity. Here $\delta_{i,j}$ are the kronecker symbols and we notice $|\frac{\partial}{\partial x_i} f_j - \delta_{i,j}| < \delta$. $\qquad\square$

**Corollary B.6.** *There is a $0 < \delta < \frac{1}{d-1}$ such that for any $f$ that is $(\delta, 1)$-near-identity, $f$ can be written as $f_1 \circ f_2 \circ \cdots f_n$, $f_i \in S_c^{1,d}$ with $f_i(\boldsymbol{x}) = \left(x_1, x_2, \cdots, x_{i-1}, \tilde{f}_i(\boldsymbol{x}), x_{i+1}, \cdots, x_d\right)$ for $\boldsymbol{x} = (x_1, \cdots, x_d) \in \mathbb{R}^d$. If $f$ is in Diff$_c^k\left(\mathbb{R}^d\right)$ for some $k >= 1$, then so are $f_i$.*

*Proof.* By taking $\delta$ small enough, we can make $\tilde{\delta} = \delta / (1 - \delta)$ small enough, thus the $g$ in Thm. B.5 can be further decomposed. A simple observation shows that if $f$ already preserves some coordinates, then so does $g$. If we set $\delta_i = \frac{\delta_{i-1}}{1-\delta_{i-1}}$ and $\delta_1 = \delta < \frac{1}{n-1}$, by noticing $\frac{1}{\delta_{n-1}} = \frac{1}{\delta_1} - n + 2 > 1$, we can make $\delta_{n-1}$ small enough, thus the decomposition can always be continued until all the coordinates have been decomposed. $\square$

*Proof of Thm. 3.2.* It can be immediately proved by Lem. B.4 and Cor. B.6 . $\square$

## B.2 Definition of dist$_{\text{Diff}_c^1(\mathbb{R}^d)}$ and proof of Theorem 3.3

**Definition B.7.** *For any $f$ and $g \in$ Diff$_c^1(\mathbb{R}^d)$, let $\pi(f, g, \epsilon)$ denotes the set of partitions between $f$ and $g$ with maximum jump less than $\epsilon$,*

$$\pi(f, g, \epsilon) := \{(f_1, \cdots, f_n) : f_1 = f, f_n = g, \|f_i \circ f_{i+1}^{-1} - I\|_{C^k} < \epsilon, 1 \le i < n, n \in \mathbb{N}^+\}$$

*then we define*

$$dist_{Diff_c^1(\mathbb{R}^d)}(f, g) := \limsup_{\epsilon \to 0} \min_{(f_1, \cdots, f_n) \in \pi(f, g, \epsilon)} \sum_{i=1}^{n-1} \|f_i \circ f_{i+1}^{-1} - I\|_{C^k}.$$

*Proof of Thm. 3.3 .* By Lem. B.4, $f$ can be decomposed into $s_1$ many $(\frac{1}{d-1}, 1)$-near-identity diffeomorphisms with $s_1 \approx (d - 1)\ell$; by Cor. B.6, each $(\frac{1}{d-1}, 1)$-near-identity diffeomorphism can be decomposed into at most $d$ single-coordinate transforms, thus $s \le s_1 \cdot d \approx d(d-1)\ell$. The evaluation for $\ell$ is difficult, but a lower bound is straightfoward: $\ell \ge \|f - I\|_{C^1}$. When $f$ is not far from $I$, the lower bound is adequate. More details refer to [4]. $\square$

## B.3 Proof for Theorem 3.4

*Proof.* Take any positive number $1 > \tilde{\epsilon} > 0$ and compact set $K \in \mathbb{R}^d$. Put $r \triangleq \max_{\boldsymbol{x} \in K} \|f_1(\boldsymbol{x})\|$ and $K' \triangleq \{\boldsymbol{x} \in \mathbb{R}^d : \|\boldsymbol{x}\| \le r + 1\}$. Let $g_2 \in G$ satisfying

$$\|f_2 - g_2\|_{C^k(K')} < \tilde{\epsilon}.$$

Since any continuous map is uniformly continuous on a compact set, we take a positive number $\delta > 0$ such that for any $\boldsymbol{x}, \boldsymbol{y} \in K'$ with $|\boldsymbol{x} - \boldsymbol{y}| < \delta$,

$$\sup_{|\alpha| \le k} \|D^\alpha f_2(\boldsymbol{x}) - D^\alpha f_2(\boldsymbol{y})\| < \tilde{\epsilon}.$$

From the assumption, we can take $g_1 \in G$ satisfying

$$\|f_1 - g_1\|_{C^k(K)} < \min\{1, \delta\}.$$

Then it is clear that $f_1(K) \subseteq K'$ by the definition of $K'$. Moreover, we have $g_1(K) \subseteq K'$. In fact, we have

$$\|g_1(\boldsymbol{y})\| \le \sup_{\boldsymbol{x} \in K} \|f_1(\boldsymbol{x}) - g_1(\boldsymbol{x})\| + |f_1(\boldsymbol{y})| \le 1 + r$$

for any $\boldsymbol{y} \in K'$.

Then for any $\boldsymbol{x} \in K$, we have

$$\|f_2 \circ f_1 - g_2 \circ g_1\|$$
$$\le \|f_2 \circ f_1 - f_2 \circ g_1\| + \|f_2 \circ g_1 - g_2 \circ g_1\| < 2\tilde{\epsilon}.$$

Now let us consider the cumulative error for derivatives. We have

$$\|D(f_2 \circ f_1) - D(g_2 \circ g_1)\|$$
$$\leq \|D(f_2 \circ f_1) - D(f_2 \circ g_1)\| + \|D(f_2 \circ g_1) - D(g_2 \circ g_1)\|$$
$$= \|(Df_2) \circ f_1 \cdot Df_1 - (Df_2) \circ g_1 \cdot Dg_1\| + \|(Df_2) \circ g_1 \cdot Dg_1 - (Dg_2) \circ g_1 \cdot Dg_1\|$$
$$\leq \|(Df_2) \circ f_1 \cdot D(f_1 - g_1)\| + \|(D(f_2 - g_2)) \circ g_1 \cdot Dg_1\| + \|((Df_2) \circ f_1 - (Df_2) \circ g_1) \cdot Dg_1\|$$
$$< (\|Df_2\| + 2\|Dg_1\|)\,\tilde{\epsilon}$$
$$< C(f_1, f_2)\,\tilde{\epsilon}$$

by noticing that
$$\|Dg_1(\boldsymbol{x})\| \leq \|Df_1(\boldsymbol{x})\| + \tilde{\epsilon} \leq \|Df_1(\boldsymbol{x})\| + 1.$$

Higher order derivatives can be estimated following the same procedure with more complex computations and reusing of triangular inequality. We can finally arrive at
$$\|f_2 \circ f_1(\boldsymbol{x}) - g_2 \circ g_1(\boldsymbol{x})\|_{C^k(K)} < \tilde{C}(f_1, f_2)\,\tilde{\epsilon}$$

with $\tilde{C}(f_1, f_2)$ only depends on $f_1$ and $f_2$ and their derivatives, doesn't depend on $\tilde{\epsilon}$ because $f_1, f_2$ are compactly supported, which means they have finite high order derivatives over $\mathbb{R}^d$.

Thus we take $\tilde{\epsilon} = \frac{\epsilon}{\tilde{C}(f_1,f_2)}$, then $\|f_2 \circ f_1(\boldsymbol{x}) - g_2 \circ g_1(\boldsymbol{x})\|_{C^k(K)} \leq \epsilon$, and then finished our proof. $\qquad\square$

## B.4 Proof for Theorem 3.6

*Proof.* Note that
$$\iota_d \circ \tau \circ \pi_d(x_1, x_2, \cdots, x_d, 0) = (x_1, x_2, \cdots, \tau_d(\boldsymbol{x}), 0).$$

This can be decomposed into three small steps:

$$(x_1, x_2, \cdots, x_d, 0) \xrightarrow{\phi_1} (x_1, x_2, \cdots, x_d, \tau_d(\boldsymbol{x})) \xrightarrow{\phi_2} (x_1, x_2, \cdots, \tau_d(\boldsymbol{x}), x_d) \xrightarrow{\phi_3} (x_1, x_2, \cdots, \tau_d(\boldsymbol{x}), 0).$$

Next let us approximate $\phi_1, \phi_2, \phi_3$ using the elements in $\mathcal{G}$-INN$_{d+1}$. By definition, $\phi_1$ can be written as $\Phi_{d+1,d,\sigma,t}$ with $\sigma$ to be any function, $t(\boldsymbol{x}) = \tau_d(\boldsymbol{x})$. By assumption, $\mathcal{H}$ has $C^k$-universality for $t$, thus we know $\mathcal{G}$-INN$_{d+1}$ has universality for $\phi_1$. $\phi_2$ is just a permutation which is already in our layers. $\phi_3$ can be written as $\Phi_{d+1,d,\sigma,t}$ with $\sigma = 0$, $t(\boldsymbol{x}) = -\tau_d^{-1}(\boldsymbol{x})$. Here $\tau_d^{-1}(\boldsymbol{x})$ is the inverse of $\tau_d(\boldsymbol{x})$ w.r.t. $x_d$ because $\tau_d(\boldsymbol{x})$ is a monotonic function w.r.t. $x_d$. Thus, we claim that $\mathcal{G}$-INN$_{d+1}$ has universality for $\phi_3$. By Thm. 3.4, we know that $\iota_d \circ \tau \circ \pi_d$ can be arbitrarily approximated by $\mathcal{G}$-INN$_{d+1}$.

Thus, for any $\epsilon > 0$, there exists a $\tilde{\tau} \in \mathcal{G}$-INN$_{d+1}$ such that
$$\|\iota_d \circ \tau \circ \pi_d - \tilde{\tau}\|_{C^k(K \times \mathbb{R})} < \epsilon,$$

and furthermore,
$$\|\tau - \pi_d \circ \tilde{\tau} \circ \iota_d\|_{C^k(K)}$$
$$= \|(\pi_d \circ \iota_d) \circ \tau \circ (\pi_d \circ \iota_d) - \pi_d \circ \tilde{\tau} \circ \iota_d\|_{C^k(K)}$$
$$\leq \|\pi_d\| \cdot \|\iota_d \circ \tau \circ \pi_d - \tilde{\tau}\|_{C^k(K \times \mathbb{R})} \cdot \|\iota_d\| < \epsilon.$$

$\qquad\square$

## B.5 Proofs for Theorem 3.7

**Definition B.8.** *Isotopies along Diff$_c^{k,m,d}$. An isotopy between two diffeomorphisms $\phi_0, \phi_1 \in$ Diff$_c^{k,m,d}$ is a $C^k$-map $H : [0,1] \times \mathbb{R}^{m+d} \to \mathbb{R}^{m+d}$ such that the mapping $h_t : \mathbb{R}^d \to \mathbb{R}^d$ defined by $h_t(x) = H(t,x)$ for all $t \in [0,1]$, $h_0 = \phi_0, h_1 = \phi_1$ and $h_t \in$ Diff$_c^{k,m,d}$ for all $t \in [0,1]$. It turns out that $t \to h_t$ is a continuous path in the group Diff$_c^{k,m,d}$ joining $\phi_0$ to $\phi_1$.*

**Theorem B.9.** *The group Diff$_c^{k,m,d}$ is connected. Moreover, the group of diffeomorphisms with compact supports which are isotopic to the identity map $I$ through compactly supported isotopies coincide with Diff$_c^{k,m,d}$. Here the identity map $I$ means $I(x) = x$ for all $x \in \mathbb{R}^{m+d}$.*

*Proof.* For arbitrary $f \in \text{Diff}_c^{k,m,d}$, we have a compactly supported $C^k$-isotopy

$$H : \mathbb{R}^{m+d} \times [\epsilon, 1] \to \mathbb{R}^{m+d},$$

given by the proportionately contraction

$$H\left((\boldsymbol{y}, \boldsymbol{x}), t\right) \triangleq tf\left(\boldsymbol{y}/t, \boldsymbol{x}/t\right), \boldsymbol{y} \in \mathbb{R}^m, \boldsymbol{x} \in \mathbb{R}^d.$$

It is obvious that $H\left((\boldsymbol{y}, \boldsymbol{x}), t\right)$ always lie in $\text{Diff}_c^{k,m,d}$, the first $m$-coordinate corresponding to $y$ are fixed. By choosing $\epsilon$ small enough, we can achieve $h_\epsilon\left(\boldsymbol{y}, \boldsymbol{x}\right) = H\left((\boldsymbol{y}, \boldsymbol{x}), \epsilon\right) \in \left(\text{Diff}_c^{0,m,d}\right)_0$, the $C^0$-connected neighborhood of identity $I$.

Thus, there is a prolongation of $H : \mathbb{R}^{m+d} \times [0, 1] \to \mathbb{R}^{m+d}$ that is a compactly supported $C^0$-isotopy from $I$ to $f$ with $H_t = I$, for all $t \in [0, \epsilon/2]$. Since compactly supported $C^k$-isotopy space with first $m$-coordinate fixed is dense in compactly supported $C^0$-isotopy space with first $m$-coordinate fixed, and since $H$ is already $C^k$ on $\mathbb{R}^{m+d} \times \left([0, \epsilon/2] \cup [\epsilon, 1]\right)$, we can always find a compactly supported $C^k$-isotopy $\tilde{H}$ from $I$ to $f$, and thus $f$ is isotopic to the identity $I$ in $\text{Diff}_c^{k,m,d}$. □

Follow the same proof in Lem. B.4, we can get the following lemma.

**Theorem B.10.** *For any diffeomorphism $f \in \text{Diff}_c^{k,m,d}$ and any $\delta > 0$ of the identity, there exists a finite sequence of $(\delta, k)$-near-identity diffeomorphisms $g_1, \cdots, g_s \in \text{Diff}_c^{k,m,d}$ such that $f = g_s \circ g_{s-1} \circ \cdots \circ g_1$.*

We remark that Thm. B.5 and Cor. B.6 need no modification and can be directly applied to here.

*Proof.* of Thm. 3.7.

It is immediately proved by Lem. B.10 and Cor. B.6 by replac int $d$ in Cor. B.6 with $m + d$. □

### B.6 Proofs for Theorem 4.2

*Proof.* By the famous Constant Rank Theorem [7], there exists a parametric ($\boldsymbol{c}$ plays the role of parameter) diffeomorphism $f_{\boldsymbol{c}}(\cdot) : \mathbb{R}^{d_\phi} \to \mathbb{R}^{d_\phi}$ such that

$$f_{\boldsymbol{c}}(\boldsymbol{a}, \boldsymbol{0}_{1 \times (d_\phi - d_a)}) = \Phi(\boldsymbol{c}, \boldsymbol{a}), \text{ for all } (\boldsymbol{c}, \boldsymbol{a}) \in \mathcal{C} \times \mathcal{A}.$$

Then we can approximate $f_{\boldsymbol{c}}$ by Thm. 3.6, 3.8, given that $d' = \max\{d_a, [\frac{d}{2}]\} \geq d_\phi$. □

## C   Learning invertible map under dimension augmentation

In this section, we analysis how to make the affine coupling flow with dimension-augmentation invertible. For simplicity we only consider the non-parametric case. The input is $\boldsymbol{x} \in \mathbb{R}^d$, and after dimension-augmentation $\boldsymbol{x} \to (\boldsymbol{x}, \boldsymbol{0})$ and some invertible map layers $f$, output is $F(\boldsymbol{x}) = f(\boldsymbol{x}, \boldsymbol{0}) \in \mathbb{R}^{d+r}$, where $r$ is the augmented dimension. Note that and $F$ is not a surjective map to $\mathbb{R}^{d+r}$, and $\text{Range}(F) \in \mathbb{R}^{d+r}$ is a manifold of dimension $\mathbb{R}^d$. Given any $\boldsymbol{y} \in \mathbb{R}^{d+r}$, it's not always true that we can find $\boldsymbol{x} \in \mathbb{R}^d$ such that $F(\boldsymbol{x}) = \boldsymbol{y}$ unless $\boldsymbol{y} \in \text{Range}(F)$.

To handle this problem, we need to make sure that $\text{Range}(F)$ is tractable for easy sampling. The easiest way is to make $\text{Range}(F) \approx \mathbb{R}^d \times \{0\}^r$. Note that in this way, for any $\boldsymbol{y} \in \mathbb{R}^d$, we can always find $\boldsymbol{x} \in \mathbb{R}^d$ such that $f(\boldsymbol{x}, \boldsymbol{0}) = f^{-1} = (\boldsymbol{y}, \boldsymbol{0})$, and note that $f$ is an invertible map, which means $(\boldsymbol{x}, \boldsymbol{0}) = f^{-1}(\boldsymbol{y}, \boldsymbol{0})$. Such a $f$, by Thm. 3.6, Cor. 3.5 and Thm. 3.3, can be approximated arbitrary well in $C^k$ norm by a fixed-layer-number affine-coupling flow $\hat{f}$.

To summarize, if we want to learn an invertible map from $\boldsymbol{x} \in \mathbb{R}^d$ to $\boldsymbol{y} \in \mathbb{R}^d$, we first augment $\boldsymbol{x}$ and $\boldsymbol{y}$ to $(\boldsymbol{x}, 0) \in \mathbb{R}^{d+r}$ and $(\boldsymbol{y}, 0) \in \mathbb{R}^{d+r}$; then apply affine-coupling flow to learn the map $\hat{f}$ from $(\boldsymbol{x}, 0)$ to $(\boldsymbol{y}, 0)$. After that, we apply $\hat{f}^{-1}$ to $(\boldsymbol{y}, 0)$, and get the initial-domain samples.

# D  Experiments

## D.1  Taiji

In this section we show the learning results of `Para-CFlows` on the Taiji tasks. The experimental setting is stated in Section 5.1. We compare the effects of hidden dimensions $d_{hid}$ and number of layers $N_{layer}$. As we see, when $d_{hid} = 2, N_{layer} = 6$, the learned result is far from correct; when $d_{hid} = 3, N_{layer} = 6$ or $d_{hid} = 2, N_{layer} = 9$, the results are imperfect; Other situations are almost the same. We can also refer to Fig. 3(c) for the MSE under different setting, where we can see when $d_{hid} = 5, N_{layer} = 6$, it can do better than $d_{hid} = 5, N_{layer} = 15$. Thus we believe that raising to higher dimensions can do better than stacking more layers under similar extra parameter size.

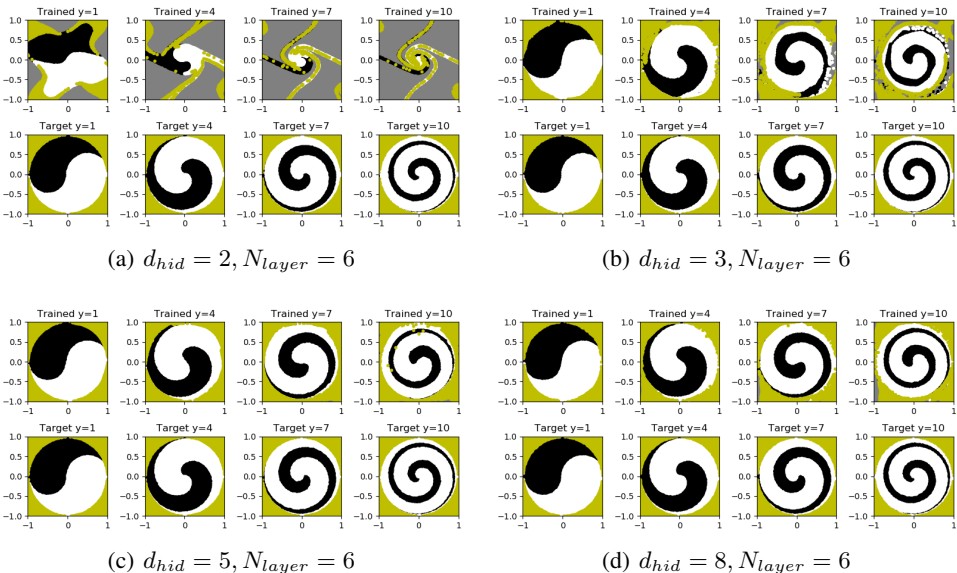

Figure 5: Different training results under different hidden dimensions

## D.2  BO test

### D.2.1  Experimental settings

In our experiments, we use 3 well-known benchmark functions: `Ackley`, `Trid` and `Rastrigin` [41]. It is noted that the original benchmark tasks are **contextless**. Here, we simply set the first $d_c$ dimensions as the context vector and leave the last one dimension as the action to be optimized. For our constructed optimization problems, `Ackley` has the action dimension that is coupled with all context dimensions, `Trid`'s action is coupled with just one context dimension, and all the dimensions of `Rastrigin` are independent with each other. To construct context-dominating tasks where the reward depends much more on the context than the action, we set $d_c \gg 1$.

In the training of all the surrogate models, we consistently set the number of batch size to 64, number of the epochs to 200, and the learning rate to 0.01. For cases when the dimensionality of context is 5, 10 and 20, we implement each of the aforementioned surrogate models in Sec. 5.2 with similar sizes (number of trainable parameters) and the corresponding specifications are posted as in Tab.2, Tab. 3, and Tab. 4, respectively.

### D.2.2  KT

To further verify the sensitiveness, we train all models with uniformly randomly sampled data (both context and action) of various sizes and then test the trained models on 10,000 testing contexts, each with the action space swept. We calculate the Kendall's Tau (KT) score [27] between $f_c(a)$ and

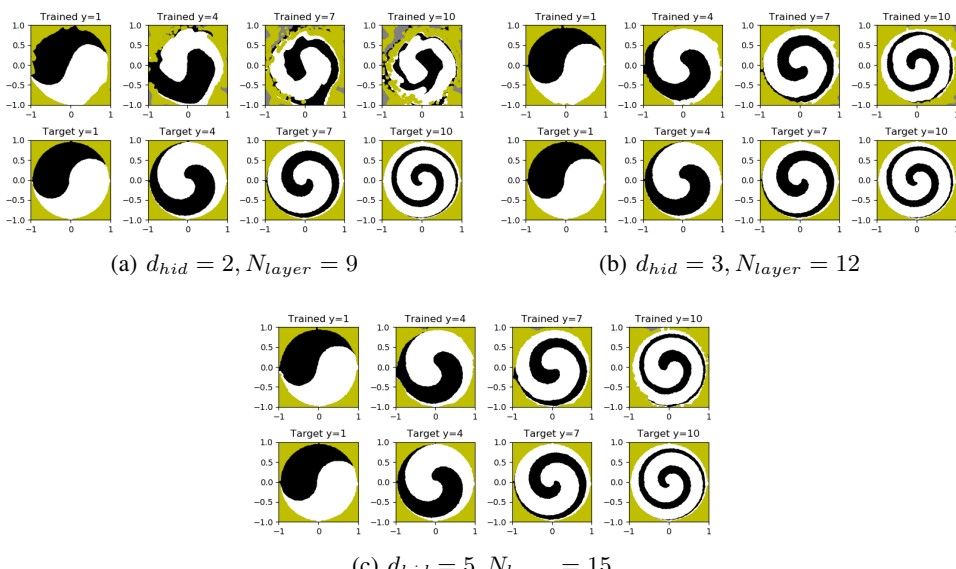

(a) $d_{hid} = 2, N_{layer} = 9$   (b) $d_{hid} = 3, N_{layer} = 12$

(c) $d_{hid} = 5, N_{layer} = 15$

Figure 6: Different training results under different number of stacking layers

Table 2: When dimensionality of context is 5: #hidden-layers and #hidden-nodes represent the number of hidden layers and nodes for state network in `Para_CFlow` and base network in other models, respectively. #flows is the number of modules in a flow-based model. #param is the total number of trainable parameters in a neural surrogate model.

| Method | #hidden-layers | #hidden-nodes | #flows | #parameters |
|---|---|---|---|---|
| Para-CFlow | 1 | 64 | 3 | 1428 |
| MLP | 2 | 32 | 0 | 1313 |
| MLP-Ascend | 2 | 32 | 0 | 1481 |
| Resnet | 2 | 32 | 0 | 1346 |

Table 3: When dimensionality of context is 10: The meaning of the header is the same as in Tab. 2.

| Method | #hidden-layers | #hidden-nodes | #flows | #parameters |
|---|---|---|---|---|
| Para-CFlow | 0 | 128 | 3 | 5337 |
| MLP | 2 | 64 | 0 | 4993 |
| MLP-Ascend | 2 | 64 | 0 | 5669 |
| Resnet | 2 | 64 | 0 | 5058 |

Table 4: When dimensionality of context is 20: The meaning of the header is the same as in Tab. 2.

| Method | #hidden-layers | #hidden-nodes | #flows | #parameters |
|---|---|---|---|---|
| Para-CFlow | 1 | 64 | 3 | 12723 |
| MLP | 3 | 64 | 0 | 9793 |
| MLP-Ascend | 3 | 64 | 0 | 11469 |
| Resnet | 3 | 64 | 0 | 9922 |

$\hat{f}_{\boldsymbol{c}}(a)$, which measures the action order consistency between the groundtruth and the prediction. As shown in Fig. 7, `Para-CFlow` exhibits higher KT scores than the other models under higher dimensional context, demonstrating that it indeed preserves critical information of the action.

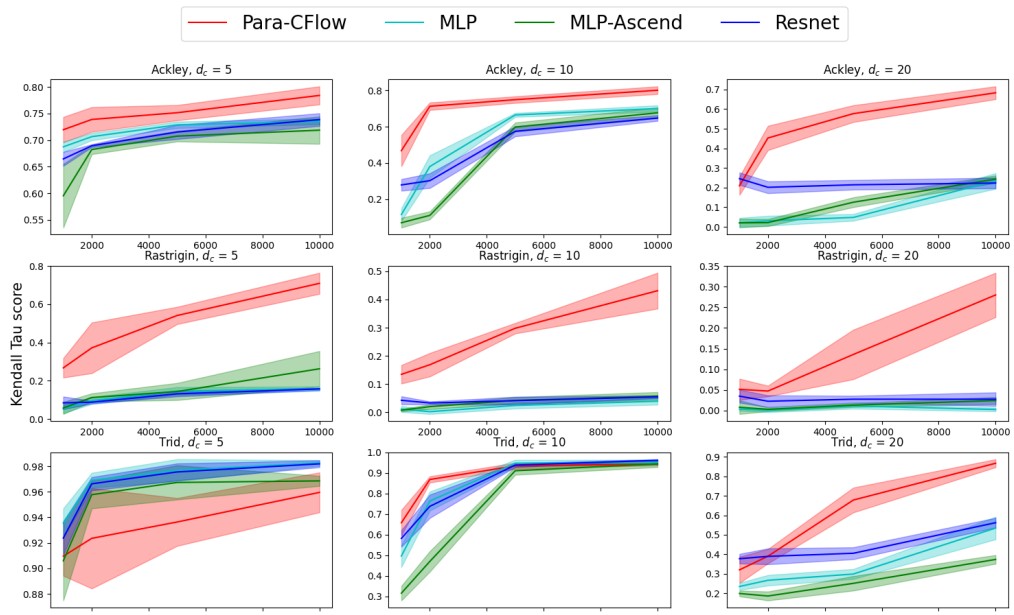

Figure 7: KT scores calculated from 5 independent trials with the context dimension as 5, 10 and 20.

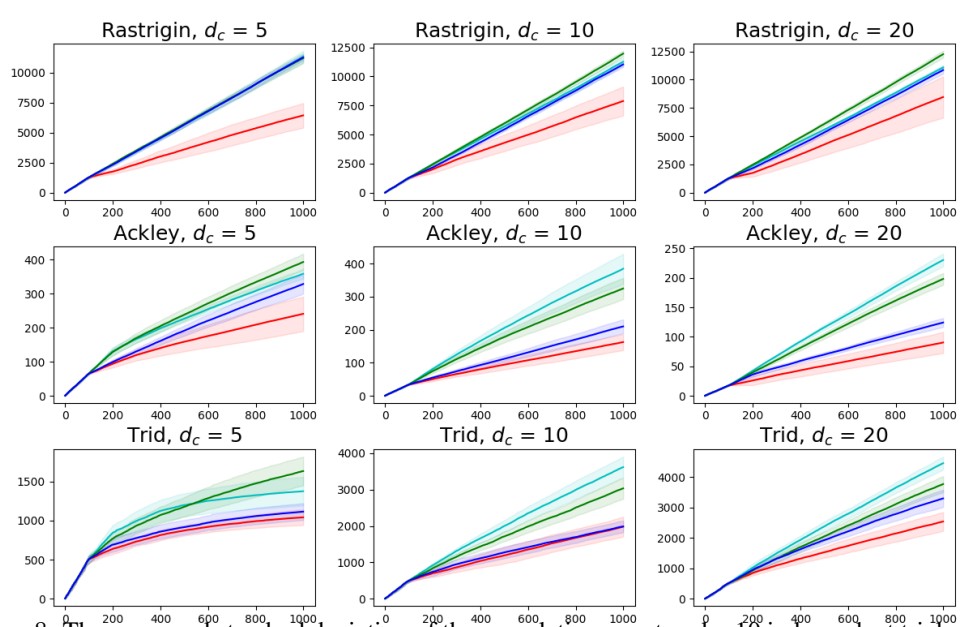

Figure 8: The mean and standard deviation of the cumulative regret under 10 independent trials using *discrete search* with different neural surrogate models for context of dimensionality 5, 10, and 20 on `Rastrigin`, `Ackley` and `Trid`, respectively.

### D.2.3 Discrete search

The cumulative regrets of BO using grid search (That is, we traversal all $a$ in `linespace`$(-3, 3, 100)$ in each surrogate-prediction state) with different neural surrogate models for context of dimensionality 5, 10 and 20 are shown in Fig. 8. Comparing this result with 4(b), we can see that `Para-CFlow` has lower cumulative regrets when using Gradient Decent than grid search, but the results for other three models do not support this. Besides, even in grid search, `Para-CFlow` still does the best among different models, which means it preserves better action sensitivities.