# OpenReview forum: "Para-CFlows: $C^k$-universal diffeomorphism approximators as superior neural surrogates"
_NeurIPS.cc/2022/Conference — NeurIPS 2022 Accept_

### Official Review · Reviewer_hkZK · 2022-07-11

**Rating:** 7
**Confidence:** 3
**Soundness:** 3 good
**Presentation:** 3 good
**Contribution:** 3 good

**Summary:**

This paper discussed the expressive power of invertible neural networks based on coupling flows (CFlows) for approximating diffeomorphisms. Specifically, suppose a class of invertible functions can approximate a small class of diffeomorphisms consisting of single coordinate affine transformations in $C^k$-norm. In that case, this class can approximate a class of compactly-supported diffeomorphisms (Corollary 3.5 for non-the parametric case, Theorem 3.8 for the parametric case). Furthermore, This paper showed that single-coordinate affine coupling flow (SACF) with one-dimensional dimension augmentation has $C^k$-universality (Theorem 3.6).
Based on the above analysis, this paper proposed Para-CFlow, which has universality under the rank-preserving assumption. Then, this paper applied Para-CFlow to a contextual BO task to verify its usefulness empirically.

**Questions:**

- l.221: $\mathcal{A}\in \mathbb{R}^{d_a}$ -> $\mathcal{A}\subset \mathbb{R}^{d_a}$
- l.353: Te -> The

**Limitations:**

The conclusion section discussed the technical limitations of the proposed Para-CFlow. Specifically, this paper pointed out that a fast log-determinant is impossible (l.362).

**Strengths And Weaknesses:**

Strengths

- This paper extended existing theories with simpler proofs.
- The numerical experiments tasks that require function derivatives. These experiment settings align with the theory part.
- Presentation is good.

Weaknesses

- Other studies also dealt with the $C^k$-universality of invertible models.
- The proposed method, Para-CFlow, is difficult to use for applications to which we usually employ invertible models such as image generation.


Originality（Novelty）

- Recently, [1] extended Teshima et al. (2020) by developing the theory of universality of invertible models with respect to the Sobolev-norm, which includes function derivatives. However, since this paper has been recently published, I do not think that this paper significantly reduces its novelty.
  - [1] https://arxiv.org/abs/2204.07415


Quality（Soundness）

- As far as I checked, the proof is correct.
- Applying the theory and models to contextual BO tasks is appropriate,  as it requires (first-order) derivatives (l.262).

Clarity（Presentation）

- The organization of this paper is OK.
- Mathematical descriptions are good, and proofs are easy-to-follow.

Significance

- Although the architecture is different, this paper proved the $C^k$-universality of SACFs (with minimal preprocessing) that Teshima et al. (2020) did not deal with.
- This paper gave a more straightforward proof of universality than Teshima et al. (2020). It makes the theory more accessible to many researchers and deepens the understanding of the expressive power of the invertible model.

---

> ### Author Response · Authors · 2022-07-29
> **Author Response**
>
> Thank the reviewer very much for your positive comments on our strengths, significance and originality. We agree with your comment that "this paper significantly reduces its novelty".  Our work and [1] are concurrent works, and [1] was not avaliable when we started working on our work. We will include the discussion on this recently published work in our revised version if it is accepted.
>
> Besides, we thank the reviewer for pointing out some notation mistakes and weakness. We will fix them, and will extend our work to high dimensional applications including image generation etc.

---

> > ### Comment · Reviewer_hkZK · 2022-08-08
> > **Post-rebuttal comments**
> >
> > I thank the authors for the response. I expect the authors to fix the aformentioned points in the revised version. I keep my score.

---

### Official Review · Reviewer_15hV · 2022-07-12

**Rating:** 5
**Confidence:** 1
**Soundness:** 3 good
**Presentation:** 3 good
**Contribution:** 3 good

**Summary:**

The paper studies the approximation power of coupling flows in terms of the general functional universality, beyond the typical notion of distribution universality. The authors first develop theorems that show that the space of invertible neural networks generated by coupling flows have C^k-universality, assuming that we perform dimensional augmentation by adding a padding dimension. The authors then develop similar results for parametric problems. These approximation theorems are used to provide guarantees for the authors' new model Para-CFlows, which works for parametric problems and incorporates both dimension augmentation. Experimental results are used to corroborate the power of Para-CFlows in learning complex diffeomorphisms.

**Questions:**

Instead of asking questions, I will focus on giving suggestions to improve the clarity of the paper:
 - For section 3, instead of showcasing a list of partial theorems leading to the result (which is more suitable for the Appendix), I would suggest first stating one main theorem that represents the "final conclusion" that you want to get across to the reader. Afterwards, you can summarize the techniques/partial theorems that you used to prove that main theorem. This would make that section shorter (allowing for more space later on) and less dense for readers.
 - For section 4, I would suggest highlighting the model construction steps. This can either be done by indenting the three steps (e.g. using itemize), or creating a text box around these steps. I would also seriously consider moving Figure 6 in the Appendix to the main text, if you can create more space by cutting down content in section 3.

**Strengths And Weaknesses:**

__Quality and Clarity__: In general, the paper is well-written. The authors clearly introduce all the concepts that they used, and presented a detailed roadmap of how they developed their theorems (I placed some suggestions on how to organize them in the Questions part). The experiments are also comprehensive and convincing.

__Originality and Significance__: Unfortunately, I work more on normalizing flows for learning distributions, so the subject of this paper is a bit beyond my expertise, and I would refer to other reviewers to comment more on the significance. That said, I believe understanding the approximation power of coupling flows on parametric models is a relevant and strong result, and having algorithms that achieve universality would be a useful contribution to the field.

---

> ### Author Response · Authors · 2022-07-28
> **Author Response**
>
> Thank you very much for your insightful and constructive comments.
>
> We agree that the organization of section 3 and section 4 should be amended, and the page limit would not be an issue as we will have 1 extra page if the paper is accepted, we will restructure our presentation to improve the readability.

---

### Official Review · Reviewer_UuMf · 2022-07-21

**Rating:** 7
**Confidence:** 3
**Soundness:** 3 good
**Presentation:** 4 excellent
**Contribution:** 3 good

**Summary:**

This paper proves that coupling flows - in particular, flows with single coordinate affine coupling layers - can universally approximate any compactly supported diffeomorphism with non-degenerate derivative in $C^k$ norms. They also extend the result to parameteric setting, where additional parameters are to be thought of as context (and are passed to all the layers in the network without changing).

Further, they provide some experiments comparing the results to known models on synthetic datasets - to approximate known diffeomorphisms and contextual bayesian optimization tasks

**Questions:**

## Questions / Clarifications
I am very confused with proof of theorem 3.3 - it needs some more clarification. References to lemma B.4 and corollary B.6 seem to be interchanged. I am not sure that it suffices to look at the $C^1$ norm induced on $\text{Diff}_C^k$, since I believe we do want all the functions $g_i$ to be $(\delta,k)$ approximations of identity, which would require a bound on $C^k$ induced norm. Even otherwise, it is not clear to me how the bound on $s_1$ follows.

Apart from this, I found flow of the proofs to be fairly readable, and the organization of the paper was great!

## Additional Comments
1. 52: gradeint instead of gradient
2. 221, 224: confusing / inconsistent notation for function b(a,c)
3. 353: Te instead of the
4. 568: for any $f$ that is $(\delta,1)$-near-identity
5. 652: mark -> remark?

**Limitations:**

Authors have adequately addressed the limitations and potential negative societal impact of the work.

**Strengths And Weaknesses:**

## Strengths
The paper seems to be theoretically sound up to my understanding, and provides easily readable and understandable proofs for the results. The proof technique is pretty neat, and can potentially be used in other theoretical results in deep learning. It also provides more theoretical backing along the recent line of work that shows that padding helps in training coupling flows - this has been observed recently though multiple results.

## Weaknesses
1. The bound is extremely existential and does not give any sort of bounds on number of layers or parameters in terms of function properties.
2. Would be good to know how the distance function defined over $\text{Diff}_c^k(\mathbb{R}^d)$ relates to more commonly used norms of the diffeomorphism.
3. Experiments for diffeomorphism approximation are on low dimensional datasets - would be great to see how dimension affects these results.

---

> ### Author Response · Authors · 2022-07-27
> **Author response**
>
> Thank you very much for your insightful and constructive comment. We agree that our result has limitations and we provide the following clarifications.
>
> For weakness point 1 and point 2:
>
> We can define the distance over $\text{Diff}_c^k(\mathbb{R}^d)$ as follows: let $\pi(f,g, \epsilon)$ denotes the set of partitions between $f$ and $g$ with maximum jump less than $\epsilon$: $$\pi(f,g, \epsilon) := \\{ (f_1,\cdots, f_n) : f_1 = f, f_n = g, ||f_i - f\_{i+1}||\_{C^k} < \epsilon, n \in \mathbb{N} \\},$$
> then we define $$\text{dist}(f,g) := \lim\limits\_{\epsilon\to 0} \mathop{min}\limits\_{(f_1,\cdots, f_n) \in \pi(f,g, \epsilon) }  \sum\limits\_{i=1}^{n-1} ||f_i - f\_{i+1}||\_{C^k} .$$
>
> It is straight-forward to verify that this distance satisfies the triangle inequality, and $\text{dist}(f,g) \ge ||f - g||\_{C^k}$.
>
> Usually if $f$ is an arbitrary diffeomorphism, a good estimate of $\text{dist}(f,I)$ is expensive. However, if $f$ is characterized by a dynamic system, i.e. a differential equation $\frac{d f_t}{dt} = A(f_t)$ with $f_0 = I, f_T = f$, then we can then estimate   $\text{dist}(f,I)$ according to this differential equation.
>
> For Question:
>
> In Theorem B.5., we prove that if $f$ is $(\delta,1)$-near-identity, then $f$ can be decomposed into $g$ and $h$ where $g$ preserves the last coordinate, $h$ only varies  the last coordinate, and $g,h$ are $(\delta/(1-\delta),1)$-near-identity. This decomposition only needs to estimate the $C^1$ norm, and by construction if $f$ is $C^k$ then so are $g$ and $h$. Next we do the same decomposition to $g$, and repeat the same procedure. Finally we will obtain single-coordinate-transformations , and if $f$ is $C^k$ then these single-coordinate-transformations are also $C^k$.
>
>  On the other hand, estimating the $C^k$ norm of $g$ and $h$ is not an easy task, and the upper bound for $||g||\_{C^k}$ can be much larger than $\delta/(1-\delta)$, thus $\delta$ should be chosen smaller enough to guarantee further decomposition for $g$. As $(\delta,1)$-near-identity is sufficient to guarantee further decomposition, and $(\delta,k)$-near-identity implies $(\delta,1)$-near-identity ( $||\cdot||\_{C^k}\ge ||\cdot||\_{C^1} $), we choose to use the proof in our paper.
>
>
> For weakness 3:
>
> Thank you very much for your comment. We have already finished some high-dimensional experiments and will add them to the final version if hopefully accepted.

---

> > ### Comment · Reviewer_UuMf · 2022-08-07
> > **Review response**
> >
> > I am not sure that my question is answered completely. I still believe that references to Lemma B.4 and Cor B.6 do need to be interchanged (lines 578, 579).  I believe that we need $f_i$'s in the definition above are in $\text{Diff}_c^k$, the way theorem 3.3 is phrased makes it sound like they are in $\text{Diff}_c^1$ (line 164).
> >
> > I have one further question: I don't see how the bound on $s_1$ follows -- The defintion of $\ell$ ensure that there are functions $I = f_0, \ldots, f_{s_1} = f$ such that $\Vert f_i - f_{i-1} \Vert_{C^1} \le \frac{1}{d-1}$, which implies that $\Vert f_i f_{i-1}^{-1} - I \Vert_{C^1} \le \frac{1}{d-1} \Vert f_{i-1}^{-1} \Vert_{C^1}$ and this could be much larger.

---

> > > ### Author Response · Authors · 2022-08-08
> > > **Author response**
> > >
> > > Regarding the first question, thanks for pointing it out. The reference to Lemma B.4 and Cor. B.6 should be interchanged, and the expression in Theorem 3.3 is a bit confusing. We will revise it accordingly.
> > >
> > > Regarding the second question, we agree that our additional definition on $\ell$ does not make the immediate steps satisfy the original condition. It will be more starightforward to define it as
> > > $$\text{dist}(f,g) := \lim\limits\_{\epsilon\to 0} \mathop{min}\limits\_{(f_1,\cdots, f_n) \in \pi(f,g, \epsilon) }  \sum\limits\_{i=1}^{n-1} ||f_i \circ f\_{i+1}^{-1}  - I||\_{C^k} ,$$
> > >
> > > where
> > > $$\pi(f,g, \epsilon) := \\{ (f_1,\cdots, f_n) : f_1 = f, f_n = g, ||f_i \circ f\_{i+1}^{-1}  - I||\_{C^k} < \epsilon, n \in \mathbb{N} \\}.$$
> > >
> > > The limitation is that it is hard to verify if this distance is a metric. However, this definition is sufficient to state the theorem.

---

> > > > ### Comment · Reviewer_UuMf · 2022-08-09
> > > > **Reviewer response**
> > > >
> > > > Thanks for the explanation! I would expect the authors to clarify necessary definitions, and make the appropriate changes in the theorem statement, but I would keep my score.

---

### Meta-Review · Area_Chair_6Ltk · 2022-08-27

**Recommendation:** Accept
**Confidence:** Certain

**Metareview:**

The paper proves universal approximation in C^k norm for general families of invertible flows that can approximate a certain class of "entrywise" diffeomorphisms. An example of such a family are for instance, compositions of affine couplings (even entrywise) and invertible linear maps. This improves upon prior results from Teshima et al' 22 by handling arbitrary k, and the proof technique is fairly natural and clean. They also consider a certain "parametric" scenario where we want to approximate maps that are "conditional" diffeomorphisms for any fixing of some of the coordinates. Finally, they propose a new class of models, Para-CFlows which shows promising experimental results (albeit on low-dimensional, simple data).

**Award:**

No

---

### Decision · Program_Chairs · 2022-09-14

Accept